# Answering Questions by Meta-Reasoning over Multiple Chains of Thought

**Ori Yoran**[*1]     **Tomer Wolfson**[*1,2]     **Ben Bogin**[1]     **Uri Katz**[3]
**Daniel Deutch**[1]     **Jonathan Berant**[1]

[1]Tel Aviv University     [2]Allen Institute for AI     [3]Bar Ilan University

`ori.yoran@cs.tau.ac.il`          `tomerwol@mail.tau.ac.il`

## Abstract

Modern systems for multi-hop question answering (QA) typically break questions into a sequence of reasoning steps, termed *chain-of-thought* (CoT), before arriving at a final answer. Often, multiple chains are sampled and aggregated through a voting mechanism over the final answers, but the intermediate steps themselves are discarded. While such approaches improve performance, they do not consider the relations between intermediate steps across chains and do not provide a unified explanation for the predicted answer. We introduce Multi-Chain Reasoning (MCR), an approach which prompts large language models to *meta-reason* over multiple chains of thought, rather than aggregate their answers. MCR examines different reasoning chains, mixes information between them and selects the most relevant facts in generating an explanation and predicting the answer. MCR outperforms strong baselines on 7 multi-hop QA datasets. Moreover, our analysis reveals that MCR explanations exhibit high quality, enabling humans to verify its answers.

## 1   Introduction

In chain-of-thought (CoT) prompting, a large language model (Brown et al., 2020; Chowdhery et al., 2022; Kadavath et al., 2022; Touvron et al., 2023) is prompted to generate its answer following a step-by-step explanation (Wei et al., 2022; Nye et al., 2022). CoT prompting has been shown to dramatically improve performance on reasoning-heavy tasks (Kojima et al., 2022; Zhou et al., 2022). Furthermore, Wang et al. (2023) showed that sampling *multiple* chains of thought and returning their majority output further improves accuracy, a method which they term *self-consistency* (SC).

While SC leads to performance gains, it also has several shortcomings. First, when the space of possible outputs is large (Kalyan et al., 2021), each reasoning chain may lead to a different output, in

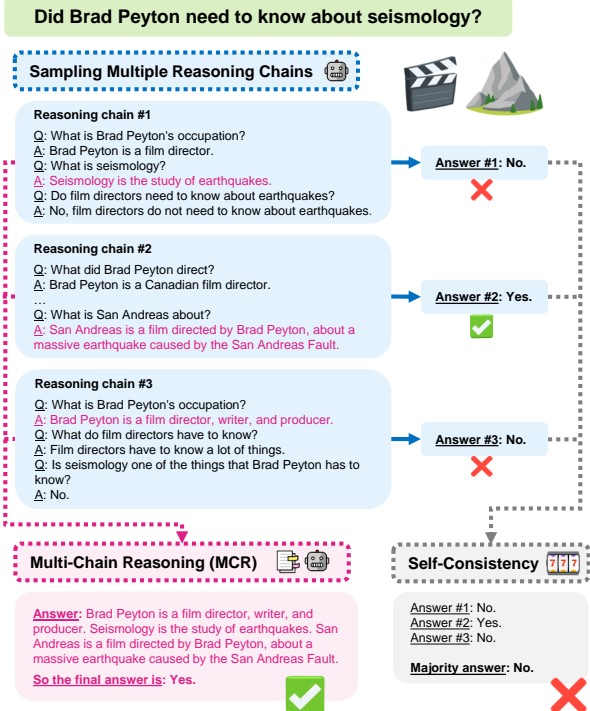

Figure 1: An example from STRATEGYQA, showing the output of Multi-Chain Reasoning versus Self-Consistency. MCR uses reasoning chains as its *context* for QA. SC solely relies on the chains' answers.

which case no significant majority will be formed. Second, focusing exclusively on the final output discards relevant information that is present in the intermediate reasoning steps. Consider answering the question *"Did Brad Peyton need to know about seismology?"* (Fig. 1). Reasoning chain #1 leads to an incorrect answer (*"No"*), but its steps provide useful information. For example, the intermediate question, and following answer, on *"What is seismology?"* constitute an important fact that is absent from the other two chains. Last, using SC jointly with chain-of-thought prompting reduces interpretability, as there is no single reasoning chain that can be considered as an explanation.

In this work, we propose *Multi-Chain Reasoning* (MCR), where we prompt a large language model

---

[*]Both authors contributed equally to this work.

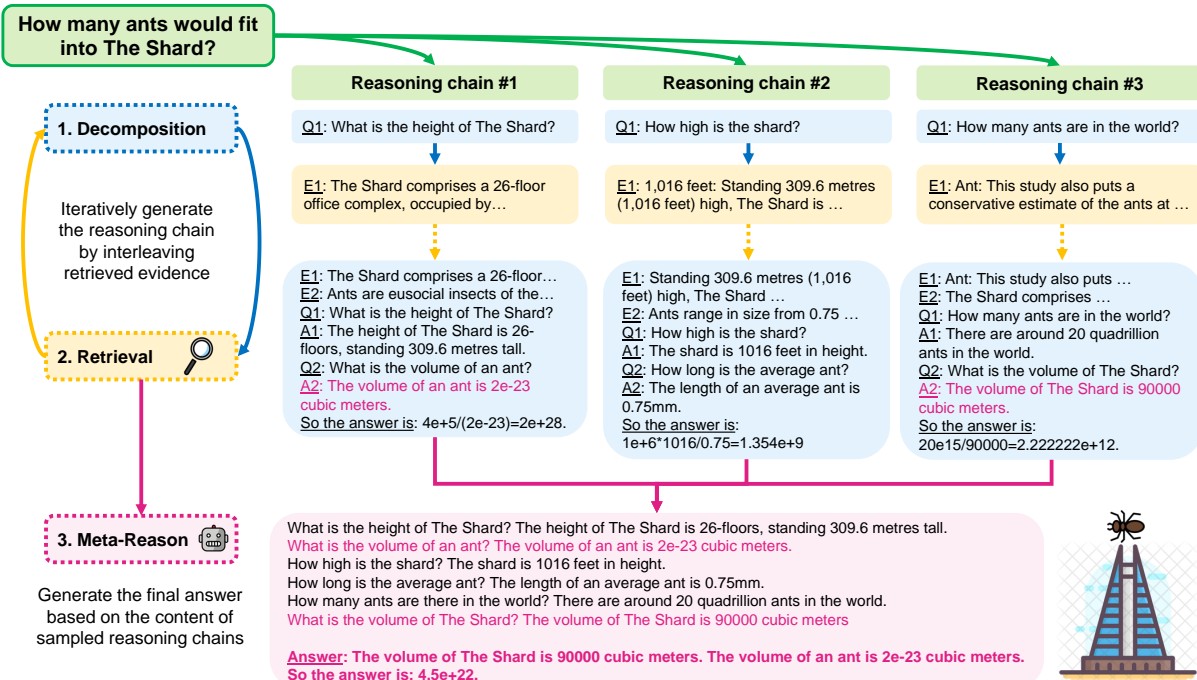

Figure 2: An overview of MCR, given a question from the FERMI dataset. Steps 1-2 generate multiple reasoning chains by conditioning the generation of intermediate questions and answers on retrieved evidence sentences. In step 3, the *meta-reasoner* generates the final answer, given multiple reasoning chains from the previous steps.

(LLM) to *meta-reason* across multiple reasoning chains and produce a final answer, alongside an explanation. Unlike prior work, sampled reasoning chains are used *not* for their predictions (as in SC) but as a means to *collect pieces of evidence* from multiple chains. Fig. 1 illustrates MCR compared to SC. While both methods rely on sampling multiple reasoning chains, SC returns the majority answer, *"No"* (grey box, bottom right). By contrast, MCR concatenates the intermediate steps from each chain (blue boxes, top left) into a unified context, which is passed, along with the original question, to a *meta-reasoner* model. The meta-reasoner is a separate LLM, prompted to meta-reason on multiple reasoning chains and produce a final answer along with an explanation (pink box, bottom left). By reasoning on multiple reasoning chains, MCR is able to mitigate the aforementioned drawbacks – it combines facts from multiple chains to produce the correct final answer, with an explanation of the answer's validity.

MCR has three main components (§3). To generate reasoning chains we use two components, a *decomposition* model and a *retriever* which jointly generate the chain (Fig. 2), similar to prior work (Press et al., 2022; Trivedi et al., 2022a). These chains are then concatenated into a unified *multi-chain context* which is fed to the aforementioned

meta-reasoner. Fig. 1 highlights the ability of the meta-reasoner to combine facts from different reasoning chains (intermediate answers in pink). The output explanation combines facts from each of the three chains: (1) *"Seismology is the study of earthquakes"*; (2) *"San Andreas is a film..."*; (3) *"Brad Peyton is a film director, writer..."*. SC (in grey) errs due to only using the answers, while the meta-reasoner reads entire reasoning chains, and is able to correctly answer the question.

We evaluate MCR on a wide range of challenging multi-hop question answering (QA) datasets, in an open-domain setting. The datasets can be categorized into two types of tasks: *implicit reasoning* tasks, where reasoning steps are implicit given the question text and need to be inferred using a strategy (Tafjord et al., 2019; Geva et al., 2021; Kalyan et al., 2021); *explicit reasoning* tasks, where a single reasoning strategy exists and can be directly inferred given the language of the question (Yang et al., 2018; Welbl et al., 2018; Press et al., 2022; Aly et al., 2021). As our baselines, we compare MCR to SC, as well as to variants of Self-Ask (Press et al., 2022) and CoT augmented with retrieval, following Trivedi et al. (2022a). Our results show MCR consistently outperforms all other baselines, in particular, beating SC by up to 5.7%, while using the same reasoning chains (§4).

We analyze the qualities of MCR in §5, by manually scoring its generated explanations and estimating their accuracy. Our analysis shows that MCR generates high quality explanations for over 82% of examples, while fewer than 3% are unhelpful. To conclude, our main contributions are:

- We introduce the MCR method for meta-reasoning on multiple chains-of-thought.
- We show that MCR outperforms all baselines, including self-consistency, on all 7 multi-hop open-domain QA benchmarks.
- We analyze MCR for its explanation quality and its multi-chain reasoning capabilities.

Our data and codebase are publicly available.[1]

## 2 Background

Recently, there has been a surge of interest in answering multi-hop questions through few-shot prompting of LLMs (Wei et al., 2022; Nye et al., 2022; Yao et al., 2022). The majority of these works follow a common standard: First, given a question, plan a step-by-step reasoning chain to derive the answer and solve all intermediate steps, aided by a retriever to minimize model hallucination (Khot et al., 2023; Press et al., 2022; Yao et al., 2022; Lazaridou et al., 2023; Trivedi et al., 2022a; Khattab et al., 2022). Then, incorporate multiple reasoning chains with answers to derive the final answer (Wang et al., 2023; Li et al., 2022). In our work, we follow this template and focus on the latter part. However, our *meta-reasoning* approach differs from prior work by reasoning on multiple reasoning chains. Namely, we use multiple chains to collect relevant evidence for question answering.

## 3 Method

We present a method for answering questions by meta-reasoning on multiple reasoning chains. Our focus is on open-domain QA, where the input is a question $q$, and the evidence to answer it is found in one or more sentences in a corpus $C$. When answering $q$ requires multiple reasoning steps, it can be expressed by a *reasoning chain*, denoted by $r$. The reasoning chain is a list of one or more intermediate question-evidence-answer triples $(q_i, e_i, a_i)$. Evidence $e_i \in C$ is a sentence that is relevant to answering the intermediate question $q_i$.

Fig. 2 describes our approach when answering *"How many ants would fit into The Shard?"*. First,

[1] https://github.com/oriyor/reasoning-on-cots

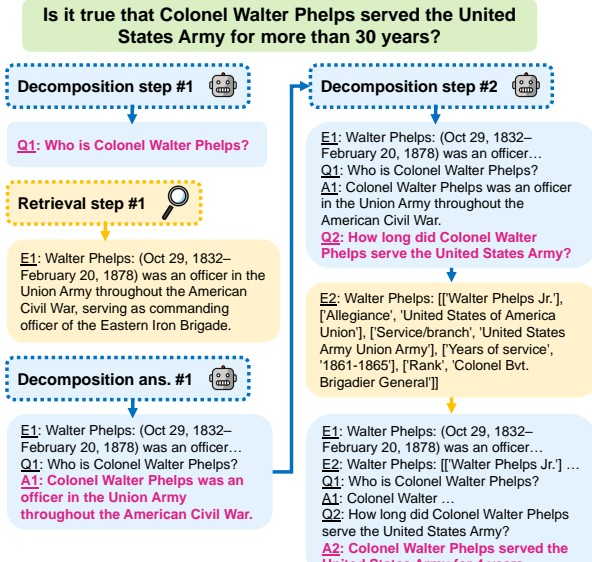

Figure 3: Interleaving decomposition and retrieval steps.

we use a prompted LLM to generate multiple reasoning chains, $r^{(1)}, ..., r^{(k)}$ (steps 1-2). Each $r^{(j)}$ is generated by interleaving generated intermediate questions with retrieved contexts (§3.1). Our main contribution is step 3: We introduce a second LLM that is prompted to *meta-reason* on multiple reasoning chains, collecting evidence facts as its explanation and generating the final answer (§3.2).

### 3.1 Generating Reasoning Chains

Given a question $q$, we generate its reasoning chain using: (1) a decomposition model, and (2) a retriever component. Our reasoning chain generation process is largely based on prior work (Press et al., 2022; Trivedi et al., 2022a), discussed in §2. Fig. 3 describes the interleaving of decomposition and retrieval. At each step, the decomposition model generates an intermediate question $q_i$, based on the original question $q$ and the previous reasoning steps. Then, the retriever uses $q_i$ to retrieve relevant evidence $e_i \in C$. We feed $e_i$ and $q_i$ to the decomposition model (along with the previous steps) to generate intermediate answer $a_i$. During answer generation, we prepend intermediate evidence sentences to the beginning of the chain rather than interleaving them, as it improves the accuracy for all baselines. For decomposition prompts, see §D.

### 3.2 Reasoning over Reasoning Chains

The meta-reasoner module is the core contribution of MCR. Instead of sampling multiple chains for their predicted *answers* (Wang et al., 2023), we utilize them for *context generation*. This context

is fed to a prompted LLM to *read* the generated chains and *reason* over them to return the answer.

In §3.1, we defined a reasoning chain as a list of $(q_i, e_i, a_i)$ triples. We first sample multiple chains and use all of their intermediate question-answer pairs $(q_i, a_i)$ as our *multi-chain context* (a variant using question-evidence pairs $(q_i, e_i)$ is described in §B.4). Fig. 2 presents the multi-chain context of the three sampled chains (lower pink box). Next, the multi-chain context and the original question are input to the meta-reasoner. This model is an LLM, few-shot prompted for QA over a multi-chain context. Fig. 4 presents one exemplar from the meta-reasoner prompt for the FEVEROUS dataset (full prompts in §D). We instruct the LLM to *"answer the question step-by-step"* given its multi-chain context, where each line describes a $(q_i, a_i)$ pair from one of the sampled chains. Next, we append the question and a step-by-step reasoning chain followed by the final answer. This last chain serves as the explanation for solving the question. The meta-reasoner is prompted with 6-10 exemplars, based on the dataset (§4.1).

Providing the meta-reasoner with multiple chains allows it to combine and aggregate facts across chains. Moreover, the model needs to extract the most relevant facts in the chains to serve as its explanation. This enables MCR to be both more accurate and more interpretable than past multi-chain approaches (as we analyze in §5).

## 4 Experiments

We compare MCR to existing methods on 7 multi-hop QA benchmarks. These cover a wide range of reasoning skills, including commonsense, composition, comparison and fact verification. MCR consistently outperforms existing approaches on all benchmarks, when experimenting with two different LLMs and retrievers. Our setting is described in §4.1 and we discuss our main results in §4.2.

### 4.1 Experimental Setting

#### 4.1.1 Datasets

As our focus is on multi-hop questions (in an open-domain setting), all datasets require *multiple* reasoning steps. Following prior work (Khattab et al., 2022; Trivedi et al., 2022a) and to limit the cost of model API calls, we evaluate on 500-1000 random examples from the development set of each

---

*Given a question and a context, answer the question step-by-step. If you are unsure, answer Unknown.*

**Context:**
Who is Robert Broderip? Robert Broderip was an English organist and composer.
Where did Robert Broderip live all his life? Robert Broderip lived in Bristol all his life.
When did Robert Broderip live? Robert Broderip lived during the 19th century.
...
Where did Robert Broderip live? Broderip lived in Bristol. During what part of the nineteenth century did Robert Broderip write music? Robert Broderip wrote music during the latter part of the eighteenth century.

**Question:** Is it true that Robert Broderip lived in London all his life and wrote a considerable quantity of music during the earlier part of the nineteenth century?

**Answer:** Robert Broderip lived in Bristol all his life, not in London. **So the answer is:** No.

Figure 4: An exemplar from the meta-reasoner prompt.

dataset.[2] We also evaluate on the official test sets of STRATEGYQA and FERMI, as they target implicit reasoning, have multiple valid strategies, and their test set evaluation cost is reasonable. For all datasets, we make sure that no evaluation questions appear in any of our prompts. Tab. 1 has example questions from each dataset. Our multi-hop QA benchmarks can be categorized based on their required reasoning skills:

- **Implicit Reasoning:** Questions that entail implicit reasoning steps (Geva et al., 2021). The reasoning steps for solving it cannot be explicitly derived from the language of the question and require commonsense or arithmetic reasoning. Such questions may have multiple valid reasoning chains. We evaluate on: STRATEGYQA (Geva et al., 2021), FERMI (Kalyan et al., 2021) and QUARTZ (Tafjord et al., 2019).
- **Explicit Reasoning:** Multi-hop questions where the reasoning steps are explicitly expressed in the language of the question (composition, comparison). These include HOTPOTQA (Yang et al., 2018), 2WIKIMQA (Welbl et al., 2018) and BAMBOOGLE (Press et al., 2022). We also evaluate on FEVEROUS (Aly et al., 2021), a fact verification dataset where claims require verifying multiple facts, and evidence may be either in sentences, tables or both.

For evaluation, we use $F_1$ to compare predicted and gold answers for all explicit reasoning datasets

---

[2]We use the entire development set for QUARTZ and BAMBOOGLE, since they include less than 500 examples. For FERMI we use all 286 "Real Fermi Problems" in its train and development sets. Exact numbers are listed in Tab. 2.

| Dataset | Example |
|---|---|
| STRATEGYQA (implicit) | Can Arnold Schwarzenegger deadlift an adult Black rhinoceros? |
| FERMI (implicit) | How many high fives has LeBron James given/received? |
| QUARTZ (implicit) | Jeff drained his rice field in the wintertime. The field likely will produce ___ crops when he uses it. A. more B. less |
| HOTPOTQA (explicit) | What city did the musician whose debut album shares its title with the 1959 Alfred Hitchcock film hail from? |
| 2WIKIMQA (explicit) | Where was the place of death of Isabella of Bourbon's father? |
| BAMBOOGLE (explicit) | What is the maximum airspeed (in km/h) of the third fastest bird? |
| FEVEROUS (explicit) | Is it true that Robert Broderip lived in London all his life and wrote a considerable quantity of music during the earlier part of the nineteenth century? |

Table 1: The multi-hop QA datasets in our experiments.

and exact-match for the binary-choice datasets. In FERMI, we use the official order-of-magnitude evaluation by Kalyan et al. (2021). We provide additional technical details on evaluation in §A.

### 4.1.2 Models

Our main models and baselines are all retrieval-augmented instances of `code-davinci-002`, prompted with in-context learning exemplars (Brown et al., 2020). In §4.3, we include additional experiments with the open-source `Vicuna-13B` (Chiang et al., 2023) LLM. Prompt exemplars are formatted as described in §3.2. The number of exemplars varies from 6-12 between datasets. Decomposition prompt exemplars are based on random examples from the train and development sets, coupled with their gold reasoning chain. For the meta-reasoner exemplars, we use reasoning chains sampled from the decomposition model as the multi-chain context. We ensure that the answer can be inferred using the sampled chains and add an explanation before the final answer, as shown in Fig. 4. For the binary-choice datasets, STRATEGYQA, QUARTZ, and FEVEROUS, the prompt contains an equal number of exemplars from each label. For additional details regarding the full prompts, length statistics and robustness to a different choice of prompts, please refer to §D.

**Meta-Reasoner** We experiment with two variants of the meta-reasoner to measure the effect of

reasoning on more than a single chain.
- **MCR:** The meta-reasoner is given five reasoning chains as its multi-chain context (§3.2). We decode one chain with greedy decoding, and sample another four reasoning chains with temperature $t = 0.7$.[3] This enables the meta-reasoner to review different pieces of evidence when answering the full question (§5).
- **SCR:** Single-Chain Reasoning (SCR) serves as an ablation for the effect of the multi-chain context. In SCR, the meta-reasoner is given the same prompt as MCR aside from having only the greedy-decoded chain in its context. This disentangles the effect of using multiple chains from the effect of having an LLM that is separate from the decomposition model to generate the final answer.

**Baselines** We evaluate the following baselines:
- **SA:** Self-Ask (Press et al., 2022) returns the answer of a single reasoning chain, that was generated with greedy decoding.
- **SC:** Self-Consistency serves as a baseline which incorporates multiple reasoning chains (Wang et al., 2023). It returns the majority answer based on multiple chains sampled from the decomposition model. We experiment with variants of 3, 5 and 15 sampled chains (SC@3, SC@5 and SC@15), in line with prior work (Wang et al., 2023; Khattab et al., 2022; Sun et al., 2023). As in MCR, we use the chain generated with greedy decoding along with additional chains sampled with $t = 0.7$.

**Retrieval** Similar to Press et al. (2022); Lazaridou et al. (2023); Paranjape et al. (2023), our models and baselines use a retriever based on Google Search, via the SerpAPI service.[4] However, we also include results using an open-source retriever (Khattab and Zaharia, 2020) in §4.3. As most of our datasets contain evidence from Wikipedia (§4.1.1), we consider it as our retrieval corpus. Therefore, we format search queries as "en.wikipedia.org $q_i$", with the Wikipedia domain preceding the intermediate question. We return the top-1 evidence retrieved by Google. Retrieved evidence may be either sentences or parsed lists. Following Trivedi et al. (2022a), we also retrieve evidence for the original question $q$. Last, all retrieved evidence sentences are prepended to the decomposition (§3.1).

---

[3]Like Wang et al. (2023), we observe that greedy-decoded chains have higher accuracy compared to the other chains.
[4]https://serpapi.com/

| Dataset | Reasoning | Examples | Oracle | SA | SC@3 | SC@5 | SCR | MCR |
|---|---|---|---|---|---|---|---|---|
| STRATEGYQA | | 1,000 | 94.4±0.1 | 69.3±0.3 | 71.5±0.8 | 72.2±0.8 | 70.0±0.6 | **73.6±0.7** |
| FERMI | implicit | 286 | 65.1±0.8 | 38.3±0.7 | 38.4±0.7 | 38.3±0.8 | 38.1±0.8 | **38.9±0.8** |
| QUARTZ | | 374 | 94.1±0.5 | 78.3±0.4 | 78.2±0.7 | 77.6±0.5 | 80.7±0.1 | **81.6±1.3** |
| HOTPOTQA | | 500 | 68.0±0.4 | 50.2±0.3 | 50.5±0.8 | 51.3±0.2 | 56.4±0.4 | **57.0±0.8** |
| 2WIKIMQA | explicit | 500 | 77.5±0.8 | 63.8±0.1 | 64.5±0.8 | 65.4±0.6 | 67.2±0.2 | **67.9±0.4** |
| BAMBOOGLE | | 120 | 77.3±0.5 | 64.6±0.6 | 64.6±0.4 | 65.0±1.5 | 64.7±0.4 | **66.5±1.7** |
| FEVEROUS | | 500 | 88.0±0.4 | 66.0±1.0 | 67.8±0.2 | 67.9±0.6 | 65.1±0.4 | **69.4±1.0** |

Table 2: Experiments using `code-davinci-002` on seven multi-hop open-domain QA datasets. Results are averaged over 3 runs. BAMBOOGLE results are averaged over 5 runs due to its smaller size.

Additional implementation details about our retrieval and MCR are described in §B.1 and §B.2.

## 4.2 Main Results

Next, we report our evaluation results. Overall, MCR outperforms our baselines on all 7 datasets.

**MCR Performance** Tab. 2 presents the results for all 7 multi-hop datasets (evaluation described in §4.1.1). We evaluate both SC@5 and MCR using five reasoning chains. In addition, we list an *oracle score* which uses the best answer out of all five chains. MCR outperforms all baselines on all of the benchmarks, beating SC@5 on STRATEGYQA (+1.4%), FERMI (+0.6%), QUARTZ (+4.0%), HOTPOTQA (+5.7%), 2WIKIMQA (+2.5%), BAMBOOGLE (+1.5%) and FEVEROUS (+1.5%).

**Adding Reasoning Chains** We measure the gains of MCR and SC when adding reasoning chains. As extending MCR is bounded by context length,[5] we follow a straightforward approach and perform self-consistency on three MCR runs. We compare this model, MCR+SC@3, which used 15 reasoning chains (5 for each MCR run), to SC@15. Tab. 3 shows that MCR+SC@3 consistently outperforms SC@15. Furthermore, though MCR uses only 5 reasoning chains, it beats SC@15 on all datasets, save STRATEGYQA. Fig. 5 plots, for each dataset, the effect that adding more reasoning chains has on meta-reasoning performance. It presents the results with 1 chain (SCR), 5 chains (MCR) and 15 reasoning chains (MCR+SC@3).

**Test Set Results** We evaluate our models on the official test sets of STRATEGYQA[6] and FERMI, which include 490 and 558 examples respectively. The results in Tab. 4 show that on STRATEGYQA MCR consistently beats SC, when using the same

---

[5]`code-davinci-002` context is capped at 8,001 tokens.
[6]https://leaderboard.allenai.org/strategyqa

| Dataset | SC@15 | MCR | MCR+SC@3 |
|---|---|---|---|
| STRATEGYQA | 74.6 | 73.6 | **76.4** |
| FERMI | 38.6 | 38.9 | **39.2** |
| QUARTZ | 78.3 | 81.6 | **82.6** |
| HOTPOTQA | 54.1 | 57.0 | **59.2** |
| 2WIKIMQA | 65.8 | 67.9 | **68.6** |
| BAMBOOGLE | 65.6 | **66.5** | 66.3 |
| FEVEROUS | 68.6 | 69.4 | **71.5** |

Table 3: Running SC and MCR on 15 reasoning chains.

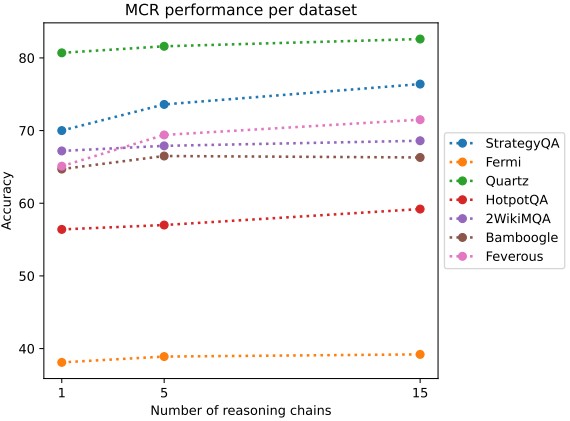

Figure 5: Per-dataset performance as a function of the number of reasoning chains used by MCR (1, 5, 15).

number of reasoning chains. In FERMI, both methods perform similarly.

**Recent Approaches** Previously, we established the advantages of meta-reasoning over multiple reasoning chains. While an apples-to-apples comparison with other recent approaches is impossible due to fundamental differences in the experimental setup (see §B.3), it serves as a rough measuring stick for the robustness of MCR across different tasks. In §B.3, Tab. 8 we compare MCR to five recent CoT-based approaches for multi-hop QA. MCR performance is comparable with the best results on all datasets (shared between these works), showcasing its robustness.

| Model | # chains | STRATEGYQA | FERMI |
|---|---|---|---|
| SC@5 | 5 | 71.4 | **39.8** |
| MCR | 5 | **72.5** | 39.7 |
| SC@15 | 15 | 74.1 | 39.7 |
| MCR+SC@3 | 15 | **75.3** | **40.1** |

Table 4: Test set results for STRATEGYQA and FERMI.

## 4.3 Open-source Models

To further examine MCR's performance (§4.2) and for better reproducibility, we experiment with an additional open-source retriever and LLM. As our retriever, we use ColBERTv2 (Santhanam et al., 2022) over the 2018 Wikipedia dump from Karpukhin et al. (2020). In addition to `code-davinci-002`, we experiment with `Vicuna-13B` (Chiang et al., 2023), a 13-billion parameters model shown to outperform LLMs like LLaMA and Alpaca (Touvron et al., 2023; Taori et al., 2023). We use the same prompts as in `code-davinci-002`, trimmed to fit a 2,048 tokens context length.

We report the full results of the open-source ColBERTv2 retriever with `code-davinci-002` and `Vicuna-13B` in Tab. 5. In addition, we provide results of open-source models when reasoning over 15 reasoning chains in Tab. 6. For `code-davinci-002`, substituting Google Search with ColBERTv2 exhibits the same trend as in Tab. 2, albeit a slight decrease in performance. MCR outperforms all other baselines, beating SC@5 on STRATEGYQA (+2.3%), FERMI (+3.4%), QUARTZ (+3.9%), HOTPOTQA (+3.5%), 2WIKIMQA (+1.2%), BAMBOOGLE (+3.6%) and FEVEROUS (+1.4%). Unsurprisingly, results sharply decrease when evaluating the smaller `Vicuna-13B` with ColBERTv2. The comparison between MCR and SCR suggests that reasoning over multiple chains is a challenge for the weaker `Vicuna-13B` model. For example, it generates open-ended answers such as *"Unknown"* or *"It depends"* for over 24% of the questions in STRATEGYQA. This suggests that meta-reasoning over multiple chains has greater gains (compared to SCR) when both the decomposition model and meta-reasoner are larger LLMs.

However, even on `Vicuna-13B`, MCR still outperforms all baselines on 5 datasets and beats SC@5 on all 7 of them: STRATEGYQA (+0.5%), FERMI (+4.6%), QUARTZ (+3.6%), HOTPOTQA (+6.5%), 2WIKIMQA (+0.3%), BAMBOOGLE

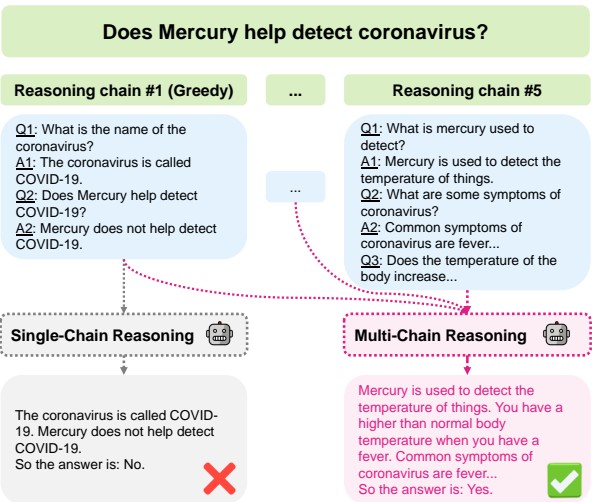

Figure 6: An example from STRATEGYQA where the greedy chain is insufficient to answer the question. MCR beats SCR by having access to multiple chains.

(+3.0%) and FEVEROUS (+1.3%). When evaluating with 15 reasoning chains, in Tab. 6, MCR+SC@3 continually beats SC@15.

## 5 Analysis

Next, we measure the importance of incorporating multiple reasoning chains in MCR and qualitatively assess its output.

**When are Multiple Chains Helpful?** In §4.2 we observed that MCR consistently outperforms single-chain reasoning (SCR). We wish to prove that this advantage lies in cases where the meta-reasoner uses additional chains. To this end, we sort examples based on the similarity of their greedy-decoded chain to the MCR explanation (details in §C.1). Lower similarity indicates less reliance of MCR on the greedy chain. Fig. 6 presents an example where the MCR explanation (pink box) includes relevant facts from a chain other than the greedy one (additional examples in §C.2). Results in Fig. 7 empirically demonstrate that on STRATEGYQA, MCR gains over SCR are highest when MCR explanations are less similar to the greedy chain. We observe this trend in all datasets (§C.1), serving as further evidence for MCR's strengths.

**Combining Reasoning Chains** In addition to choosing between reasoning chains, an interesting property of the meta-reasoner is that it can *combine* facts from different chains. We estimate the prevalence of this phenomenon on the implicit datasets, STRATEGYQA and FERMI, which are more challenging. Given an example, we automati-

| Dataset | Model | Oracle | SA | SC@3 | SC@5 | SCR | MCR |
|---|---|---|---|---|---|---|---|
| STRATEGYQA | code-davinci-002 | 94.5±0.7 | 67.1±0.6 | 69.9±0.1 | 70.8±0.6 | 67.8±0.5 | **73.1±2.1** |
| FERMI | code-davinci-002 | 64.3±0.7 | 33.2±0.3 | 33.2±0.4 | 33.1±0.4 | 33.9±0.6 | **36.5±2.1** |
| QUARTZ | code-davinci-002 | 93.9±0.6 | 77.1±0.6 | 75.6±0.7 | 76.0±1.5 | 79.3±0.3 | **79.9±1.2** |
| HOTPOTQA | code-davinci-002 | 67.7±0.7 | 50.7±0.3 | 51.5±0.6 | 52.5±0.1 | 55.3±0.2 | **56.0±1.1** |
| 2WIKIMQA | code-davinci-002 | 68.3±0.4 | 52.4±0.1 | 51.1±0.2 | 52.7±0.4 | 53.7±0.3 | **53.9±0.3** |
| BAMBOOGLE | code-davinci-002 | 56.4±1.2 | 45.9±1.1 | 47.2±1.4 | 47.0±0.7 | 47.1±1.0 | **50.6±1.3** |
| FEVEROUS | code-davinci-002 | 84.1±0.7 | 61.2±0.4 | 62.9±0.6 | 63.1±1.0 | 60.9±0.3 | **64.5±0.8** |
| STRATEGYQA | Vicuna-13B | 82.4±0.2 | 59.7±0.1 | 61.4±0.5 | 62.2±0.8 | **63.7±0.0** | 62.7±0.1 |
| FERMI | Vicuna-13B | 45.7±1.0 | 19.1±0.2 | 19.1±0.3 | 18.8±0.3 | 21.5±0.0 | **23.4±0.4** |
| QUARTZ | Vicuna-13B | 89.6±1.6 | 61.1±0.1 | 59.8±2.3 | 61.4±1.6 | 63.9±0.0 | **65.0±0.3** |
| HOTPOTQA | Vicuna-13B | 52.7±0.5 | 34.8±0.0 | 35.8±0.2 | 37.1±0.4 | 43.4±0.0 | **43.6±1.6** |
| 2WIKIMQA | Vicuna-13B | 52.2±0.3 | 32.2±0.4 | 33.8±0.6 | 34.0±1.0 | **35.1±0.0** | 34.3±0.4 |
| BAMBOOGLE | Vicuna-13B | 42.3±1.6 | 30.7±0.0 | 30.4±0.6 | 31.4±0.6 | 31.3±0.0 | **34.4±1.3** |
| FEVEROUS | Vicuna-13B | 88.7±0.2 | 61.5±0.6 | 61.0±0.6 | 61.0±0.8 | 60.6±0.0 | **62.3±1.2** |

Table 5: Experiments using ColBERTv2 retriever with code-davinci-002 and Vicuna-13B, evaluated on the development sets of our datasets. Results are averaged over 3 runs. The number of examples per dataset is the same as in Tab. 2. Vicuna-13B generation is deterministic so, using greedy decoding, SCR has a standard deviation of 0.

| Dataset | Model | SC@15 | MCR+SC@3 | Model | SC@15 | MCR+SC@3 |
|---|---|---|---|---|---|---|
| STRATEGYQA | code-davinci-002 | 72.6 | **75.6** | Vicuna-13B | 62.3 | **63.7** |
| FERMI | code-davinci-002 | 34.0 | **36.3** | Vicuna-13B | 18.8 | **23.2** |
| QUARTZ | code-davinci-002 | 76.5 | **80.7** | Vicuna-13B | 60.1 | **64.3** |
| HOTPOTQA | code-davinci-002 | 54.3 | **56.8** | Vicuna-13B | 37.8 | **44.8** |
| 2WIKIMQA | code-davinci-002 | 52.5 | **54.0** | Vicuna-13B | 35.5 | **35.6** |
| BAMBOOGLE | code-davinci-002 | 48.9 | **51.8** | Vicuna-13B | 31.8 | **35.1** |
| FEVEROUS | code-davinci-002 | 62.7 | **66.2** | Vicuna-13B | 61.1 | **64.0** |

Table 6: Running SC and MCR on 15 reasoning chains using ColBERTv2 retriever with code-davinci-002 (left columns) and Vicuna-13B (right columns).

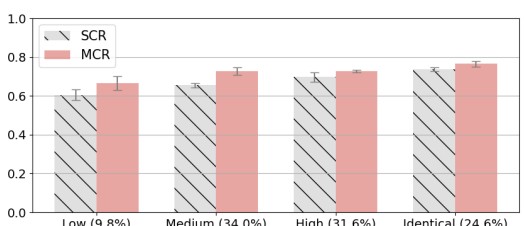

Figure 7: MCR and SCR accuracy on STRATEGYQA, categorized by the similarity of the greedy chain to the MCR explanation. When MCR uses a chain other than the greedy one (lower similarity), it outperforms SCR.

cally check if its meta-reasoner explanation is the result of combining chains. We examine if one of the output sentences appears in exactly one chain, while another sentence is absent from that chain and is part of a different chain. We consider sentences as similar if their ROUGE-1 precision is above 0.8, and distinct if it is below 0.2. Overall, in 20% of STRATEGYQA examples and 25% of FERMI, the MCR explanation results from combining reasoning chains. From a manual analysis of 50 such examples for each dataset, we observe that these multi-chain explanations are better than any individual reasoning chain in 10% of cases

(see examples in §C.2, Fig. 10). For the remaining 90%, the reasoning expressed in the resulting combination is a paraphrase of an individual chain.

**Explanation Quality** The meta-reasoner is prompted to generate an explanation alongside the final answer (§3.2). Inspired by past work (Pruthi et al., 2022), we test the quality of the MCR explanations. Four of the authors manually reviewed 600 random examples, 100 per dataset (sans FEVEROUS §B.2) and scored their meta-reasoner explanations. Each explanation is scored as either 1 (irrelevant), 2 (partially relevant) or 3 (highly relevant), based on its relevance to answering the question. We find the explanation is highly relevant in 82% of the cases (87% excluding FERMI, which is the most challenging), and is irrelevant in less than 3%.

Next, we evaluate the *faithfulness* of explanations (Jacovi and Goldberg, 2020), namely, whether a person provided only with the question and MCR explanation would answer the same as the model. Our focus was on examples with quality explanations (score 3), since they are answerable given the explanation. We answered each question based on

| Dataset | Va. | De. | Re. | Co. | Ex. | An. |
|---|---|---|---|---|---|---|
| STRATEGYQA | 20% | 24% | 8% | 15% | 20% | 17% |
| FERMI | 6% | 39% | 20% | 4% | 17% | 23% |
| QUARTZ | 14% | 6% | 13% | 19% | 11% | 40% |
| HOTPOTQA | 33% | 24% | 24% | 11% | 11% | 5% |
| 2WIKIMQA | 39% | 4% | 35% | 12% | 8% | 6% |
| BAMBOOGLE | 26% | 8% | 32% | 24% | 13% | 0% |
| FEVEROUS | 7% | 14% | 34% | 23% | 20% | 6% |

Table 7: Error classes per dataset: Valid (Va.), Decomposition (De.), Retrieval (Re.), Contradicting facts (Co.), Explanation (Ex.) and Answer (An.). We allow multiple error categories per example.

the model's explanation. In 90% of cases (95% excluding FERMI), the MCR predictions matched our own, highlighting the faithfulness of its explanations. We attribute part of the gap between human and MCR predictions to implicit reasoning tasks, where humans lead by five points, on average. For the full results, see §C.3.

**Error Analysis** We manually analyzed 700 errors by MCR (100 per dataset). We consider the following categories: *Valid* predictions where the generated answer is accurate or the original question is ambiguous; *Decomposition* errors where no chain has the necessary reasoning steps to answer the question; *Retrieval* errors where the retrieved contexts were irrelevant, leading the model to hallucinate; *Explanation* errors where MCR generates a wrong explanation while a correct one is present in the multi-chain context; *Answer* errors are when the MCR explanation is correct, but the answer is not; *Contradicting* facts are cases where MCR errs due to contrasting statements appearing in the multi-chain context.

Tab. 7 lists the prevalence of the error categories per dataset. In four datasets, over 20% of errors appear to be valid predictions, labeled as incorrect due to ambiguous questions, outdated answers or dataset errors. Decomposition is a challenge in the implicit datasets, STRATEGYQA and FERMI, with more than 24% of errors. Comparing errors on different reasoning datasets (excluding valid examples): Explanation and Answer errors are 50% on implicit reasoning datasets compared to 23% on explicit reasoning ones; Retrieval errors are more prevalent in explicit reasoning tasks with 66% of errors being due to Retrieval or Contradicting facts, compared to 30% in implicit datasets. Additional technical details on our analysis are in §C.4.

# 6    Related Work

For a thorough survey on LLM reasoning see Lu et al. (2022); Huang and Chang (2022); Qiao et al. (2022). A slew of recent works have focused on eliciting multi-step reasoning in LLMs, including scratchpads (Nye et al., 2022), chain-of-thought prompting (Wei et al., 2022; Zhou et al., 2022), learned verifiers (Cobbe et al., 2021), selection-inference (Creswell et al., 2022) and bootstrapping (Zelikman et al., 2022).

Self-consistency (Wang et al., 2023; Fu et al., 2022) selects the majority answer across multiple chains, outperforming learned verifiers and "sample-and-rank" approaches (Adiwardana et al., 2020; Freitas et al., 2020). Li et al. (2022) further improve SC by increasing chains' diversity and introducing a trained verifier. Tafjord et al. (2022) over-samples chains and verifies them using a natural language inference model on intermediate steps, while He et al. (2022) re-rank chains based on intermediate retrieved evidence. In addition, meta-reasoning is closely tied to *self-reflection* in LLMs, which is becoming increasingly important in using the LLM to review multiple strategies (Yao et al., 2023; Shinn et al., 2023; Madaan et al., 2023).

Recent works proposed revising LLM-generated texts by using retrieved sentences (Gao et al., 2022) or model-generated feedback (Madaan et al., 2023; Chen et al., 2023; Paul et al., 2023). MCR similarly reviews LLM-generated reasoning chains however, its focus is meta-reasoning on *multiple* chains.

Significant QA research has been dedicated to reasoning over multiple facts retrieved from an underlying corpus. Such tasks include multi-step questions that require explicit reasoning (Talmor and Berant, 2018; Welbl et al., 2018; Wolfson et al., 2020; Trivedi et al., 2022b), implicit reasoning (Geva et al., 2021) and multi-modal capabilities (Talmor et al., 2021).

Recent works also target retrieval-augmented LLMs, prompted to solve open-domain questions (Lazaridou et al., 2023; Khattab et al., 2022; Trivedi et al., 2022a; Ram et al., 2023; Yoran et al., 2023).

# 7    Conclusion

This work introduces MCR for meta-reasoning over multiple reasoning chains. We evaluate MCR on 7 datasets for multi-hop QA that require both implicit and explicit reasoning in an open-domain setting and show that it outperforms previous approaches on all evaluation benchmarks.

## 8 Limitations

In this work we introduce a meta-reasoner model to reason over multiple reasoning chains. While we opt for a prompted LLM as our meta-reasoner, we do not experiment with a fine-tuned meta-reasoning model. For the meta-reasoner context, we experiment with variants which include either generated QA pairs or retrieved evidence sentences. We leave further improvements to the meta-reasoner context as future work. Due to the inference costs of current state-of-the-art LLMs we evaluate on the `code-davinci-002` model, similar to prior work (Trivedi et al., 2022a; Wang et al., 2023). However, to improve the reproducibility of our work we also provide results with an open-source LLM (Chiang et al., 2023) and retriever (Khattab and Zaharia, 2020).

## Acknowledgements

We would like to thank Harsh Trivedi, Ofir Press, Mor Geva, Peter Clark and Ashish Sabharwal for their feedback and insightful comments. We thank SerpAPI for their support by granting us an academic discount. This research was partially supported by the Yandex Initiative for Machine Learning and the European Research Council (ERC) under the European Union Horizons 2020 research and innovation programme (grant ERC DELPHI 802800). This work was completed in partial fulfillment of the Ph.D. of Ori Yoran and the Ph.D. of Tomer Wolfson.

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

## A    Evaluation

### A.1    Generating Unknown as the Answer

As we prompt LLMs to generate answers, a potential outcome is for the model to *abstain* from answering the question, by generating *Unknown* as its answer. Additional cases are when the model generates an end-of-sequence token without any final answer. In the binary-choice datasets, STRATEGYQA, QUARTZ and FEVEROUS, we assign a score of 0.5 to such examples, thereby simulating a random guess. When submitting predictions to the STRATEGYQA test set, we identify cases of model abstains or null predictions beforehand. For these examples, we assign a label of either *Yes* or *No* at random. In datasets with open-ended answers, we assign a score of 0 when the predicted answer is either *Unknown* or null. To make Self-Ask a stronger baseline, when the greedy decoded chain has a null answer, we randomly choose a prediction from one of the other chains. For SC, we do not consider predictions from chains where answers are *Unknown* or null.

### A.2    FERMI

The FERMI dataset requires approximating numeric answers for open-ended questions. Example questions are shown in Tab. 1 and Fig. 2. When providing a FERMI question to our models and baselines we also add the gold answers measure units (e.g. meters, cubes, litres, etc.). While this additional

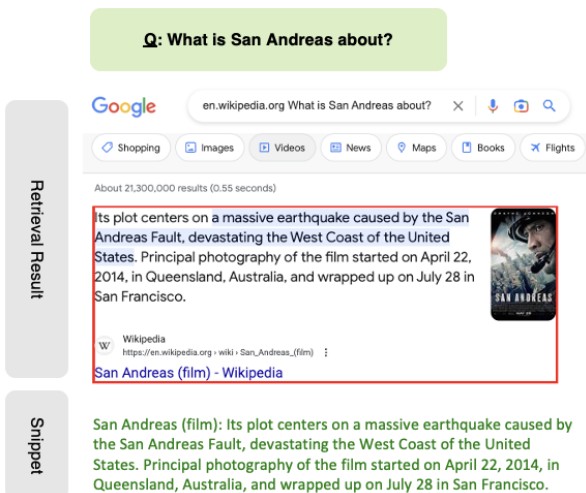

Figure 8: Example for a retrieved evidence snippet for one of the intermediate questions from Fig. 1.

input helps the model, we note that we provide it to all our baselines for a fair comparison with MCR. Nevertheless, even when given the gold units, predicting the final answers to FERMI problems remains highly challenging.

## B    Models

### B.1    Retrieval

For our retrieval, we use the Google Search Engine, via SerpAPI, and return the top-1 retrieved result as an evidence snippet. Snippets can include answer-boxes and tables.[7] We prepend the page title to the beginning of the snippet, as shown in Fig. 8.

### B.2    Implementation Details

We describe the design choices made in our MCR model, such as preforming retrieval on the original question and a variant of the meta-reasoner prompt for FEVEROUS. Due to cost limitations, we evaluate our design choices at a smaller scale and avoid running an exhaustive grid search.

**Retrieving the Original Question**    We follow past work (Trivedi et al., 2022a) by incorporating retrieved evidence for the original question in addition to evidence retrieved for the intermediate steps (§3.1). This has a positive or negligible effect on most datasets however, it dramatically decreases the results of all models on the FERMI task. Results drop for SA (38.3±0.7 to 34.7±0.5), SC (38.3±0.8 to 34.4±0.3), SCR (38.1±0.8 to 34.4±0.8) and MCR (38.9±0.8 to

---

[7]https://serpapi.com/organic-results

| Model | Ret. | LLM | # Chains | STRGYQA | HOTPOTQA | 2WIKIMQA |
|---|---|---|---|---|---|---|
| CoT (Wang et al., 2023) | no | code-davinci-002 | 1 | 73.4 | 39.8 | - |
| CoT+SC@40 (Wang et al., 2023) | no | code-davinci-002 | 40 | 79.8 | 44.6 | - |
| Self-Ask (Press et al., 2022) | yes | text-davinci-002 | 1 | - | - | 52.6 |
| DSP (Khattab et al., 2022) | yes | text-davinci-002 | 20 | - | 62.9 | - |
| IR-CoT (Trivedi et al., 2022a) | yes | code-davinci-002 | 1 | - | 61.2 | 65.2 |
| Self-Ask (ours) | yes | code-davinci-002 | 1 | 69.3 | 50.2 | 63.8 |
| MCR | yes | code-davinci-002 | 5 | 73.6 (+4.3) | 57.0 (+6.8) | 67.9 (+4.1) |
| MCR+SC@3 | yes | code-davinci-002 | 15 | 76.4 (+7.1) | 59.2 (+9.0) | 68.6 (+4.8) |

Table 8: Recent ODQA results using CoT prompting on LLMs. We list whether a retrieval component is used (Ret.), the LLM, and the number of reasoning chains used to generate the answer. Different systems evaluate on different evaluation sets for each dataset. Retrieval-augmented systems vary in terms of their retriever and corpus.

37.0±0.7). Therefore, our models are run without original question retrieval when evaluated on FERMI. Interestingly, while all models perform roughly the same without original question retrieval, MCR appears better by 2 points when evidence for the original question is used. We hypothesize that it might be due to MCR being somewhat more robust to the addition of irrelevant evidence.

**FEVEROUS Meta-Reasoner Prompt** As described in §3.2, the meta-reasoner generates an explanation which precedes the final answer. FEVEROUS is distinct from all other datasets as it require verification of multiple facts in order to verify or disprove a complex statement. When a statement is false, we list one or more of its false intermediate facts along with its correction. For example, in Fig. 4 we list that Robert Broderip lived in Bristol, not London. When prompting the meta-reasoner to list both true and false intermediate facts, we observed a decrease in performance for both MCR (69.4±1.0 to 66.4±0.7) and SCR (65.1±0.4 to 62.9±0.3). We hypothesize that repeating multiple true facts excessively prompts the model to predict the label "Yes" in cases where most (but not all) of the intermediate facts are correct.

### B.3 Empirical Comparison to Recent Approaches

In Tab. 8, we compare MCR to recent CoT-based approaches for multi-hop reasoning. An apples-to-apples comparison is not possible, as these methods do not evaluate on all 7 of our datasets and use varying samples of 500-1,000 dev examples for their evaluation. Moreover, different methods use different retrieval corpora, hyperparameters, prompts and LLMs. Nevertheless, we argue that a direct comparison serves as a measuring stick for MCR's robustness across multiple datasets, compared to

similar solutions.

Evaluation differences include the retrieval corpora, as both IR-CoT and DSP use the official Wikipedia dump provided with the HOTPOTQA dataset (Yang et al., 2018). Our retrieved evidence are from an updated version of Wikipedia, via Google Search. Since certain facts may change over time, this could potentially explain the high percentage of MCR predictions labeled as valid in our error analysis (§5).

We emphasize that our focus is on highlighting the potential of reasoning on reasoning chains. MCR is a method aimed at improving models which generate reasoning chains. Compared to SC, we observe that MCR further boosts the underlying SA model. While task-specific improvements are possible, they are orthogonal to our work.

### B.4 Reasoning on Retrieved Evidence

The meta-reasoner answers questions given a multi-chain context of *question-answer* $(q_i, a_i)$ pairs, extracted from multiple reasoning chains (§3.2). We experiment with an alternative multi-chain context, comprised of *questions* and retrieved *evidence* $(q_i, e_i)$ (§3.1). This setting resembles past work (Trivedi et al., 2022a) however, our sentences are intermediate evidence from *multiple* reasoning chains, not just the greedy-decoded chain. We compare these variants, MCR-EV and SCR-EV, to MCR and SCR that reason on QA pairs. Tab. 9 shows that meta-reasoning on retrieved evidence is less effective. The gap is more evident in implicit reasoning tasks, perhaps due to retrieved evidence being less relevant on average. Example prompts for MCR-EV and SCR-EV are listed in §D.

| Dataset | SCR-Ev | SCR | MCR-Ev | MCR |
|---|---|---|---|---|
| STRATEGYQA | 69.1±0.4 | **70.0±0.6** | 73.2±0.6 | **73.6±0.7** |
| FERMI | 34.1±0.6 | **38.1±0.8** | 33.9±0.3 | **38.9±0.8** |
| QUARTZ | 76.1±0.2 | **80.7±0.1** | 76.2±1.9 | **81.6±1.3** |
| HOTPOTQA | 53.5±0.1 | 56.4±0.4 | **58.2±1.0** | 57.0±0.8 |
| 2WIKIMQA | 66.2±0.2 | **67.2±0.2** | 67.1±0.9 | **67.9±0.4** |
| BAMBOOGLE | 64.1±0.0 | **64.7±0.4** | 67.4±2.3 | 66.5±1.7 |
| FEVEROUS | 64.0±0.5 | 65.1±0.4 | 62.5±0.5 | **69.4±1.0** |

Table 9: Effect of using question-answer pairs versus question-evidence pairs as input to the meta-reasoner.

## C  Analysis

### C.1  When are Multiple Chains Helpful?

In §5, we have shown that the advantage of MCR over SCR lies in examples where the meta-reasoner uses chains other than the one generated through greedy decoding. In Fig. 9 we provide the results for all other datasets, in addition to the STRATEGYQA results in Fig. 7. The similar trend among all datasets is that in examples with lower similarity to the greedy chain, MCR gains over SCR are higher.

The similarity between the meta-reasoner explanation and the greedy decoded reasoning chain is defined as follows: We calculate the ROUGE-1-precision (Lin, 2004) between the explanation and the chain. Low, Medium, and High are based on thresholds of $\frac{1}{3}$, $\frac{2}{3}$, and $1$ respectively, with the Identical category indicating an exact match.

### C.2  Combining Reasoning Chains

Fig. 10 provides additional examples for combining facts between multiple reasoning chains.

### C.3  Explanation Quality Analysis

We provide additional details on the annotation for the scoring meta-reasoner explanations. The annotation was performed by 4 graduate students that are authors of this paper. The annotators were presented with a question and an explanation, and asked to perform two tasks: (a) score the explanation for its quality and (b) answer the question based on the meta-reasoner explanation. We provide the full instructions shown to the annotators in Fig. 11 and the full results in Tab. 10.

### C.4  Error Analysis

We provide additional details regarding our error analysis (§5). In less than 5%, we encountered grammatically bad questions which we were unable to comprehend and were therefor discarded from our analysis. For example the HOTPOTQA

question: "What does the goddess associated with the goddess Frigg consists of what tales?"

The input to our meta-reasoner model is a context comprised of $(q_i, a_i)$ pairs, generated by the decomposition model. As the decomposition model is an LLM that is conditioned on retrieved evidence (and prior decomposition steps) it may hallucinate false intermediate answers. In cases of such hallucinations we distinguish between two error types, based on the relevant component. First, *Retrieval errors* are cases where no relevant information was retrieved, leading to the decomposition model hallucinating an incorrect $a_i$, passed on to the meta-reasoner's context. Second, we treat cases where relevant evidence was retrieved, but the decomposition model ignored it and hallucinated an incorrect $a_i$ as *Decomposition errors*.

Errors stemming from *Contradicting Facts*, are cases where the meta-reasoner context contains two contradicting facts, one accurate while the other was hallucinated by the decomposition model. For example, Fig. 12 displays an example where the context has contradicting facts on who was the father of Eliezer Ben-Yehuda. When the meta-reasoner has contradicting facts, it is expected to select the correct fact, based on the knowledge encoded in its parameters. Addressing such errors in future work could rely on refining generated text with methods such as RARR (Gao et al., 2022).

As our error classes mainly match the MCR components, this error breakdown could potentially help to guide future improvements.

## D  Prompts

### D.1  Prompt Details

We provide example prompts for our models for one explicit dataset (2WIKIMQA, decomposition: Fig. 13, MCR/SCR: Fig. 15, MCR-Ev/SCR-Ev: Fig. 17) and one implicit dataset (STRATEGYQA, decomposition: Fig. 14, MCR/SCR: Fig. 16, MCR-Ev/SCR-Ev:Fig. 18). All of our prompts

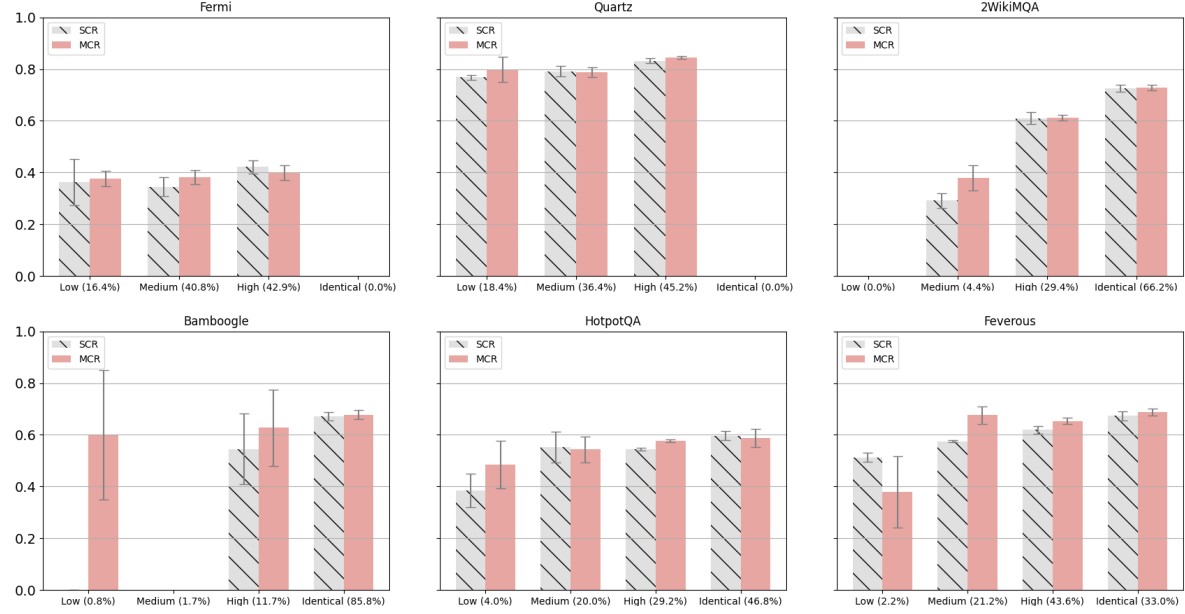

Figure 9: MCR and SCR accuracy on FERMI, QUARTZ, 2WIKIMQA, BAMBOOGLE, HOTPOTQA, and FEVEROUS, on examples categorized by their MCR explanation's similarity to the greedy chain. MCR performs similarly to SCR when similarity is high, and outperfoms SCR when similarity is lower. Error bars indicate standard deviation, which tends to be high when the number of examples in the bin is small. For FEVEROUS we display the variant where MCR has to repeat all relevant facts (§B.2), to make sure the MCR explanation is not empty.

will be released along with our codebase. We use random examples and spend minimal effort on prompt engineering. The number of exemplars varies slightly between dataset and model, with the exact numbers listed in Tab. 11.

## D.2 Prompt Statistics

In Tab. 12 we provide statistics of the sequence lengths for all of our models, which include all the decomposition prompts, output decomposition sequences, retrieved evidence and the meta-reasoning prompts. The statistics are for our decomposition model (used by all of our baselines), as well as for the meta-reasoning prompts (used by SCR and MCR). Note that generating a single reasoning chain requires multiple LLM calls, one for each decomposition step. Therefore, a single decomposition generation is generally longer than applying one additional meta-reasoning step.

Results are averaged over multiple runs, corresponding to the results in Tab. 2. Sequence lengths in Tab. 12 correspond to the number of tokens provided by the `code-davinci-002` tokenizer.

## D.3 Robustness to Choice of Prompt

We empirically measure our method's sensitivity to the prompt of choice. To this end, we randomly sampled new exemplars for both our decomposition and meta-reasoning prompts for STRATE-

GYQA and HOTPOTQA. When using different random exemplars, we observe that MCR still outperforms all baselines. Even though decomposition performance (SA) is more affected by the set of exemplars, the performance trend remains the same, with MCR being on top. Tab. 13 lists the experiment results, evaluated on 500 examples from each dataset. We also provide the original prompt results in parenthesis (averaged over 3 runs).

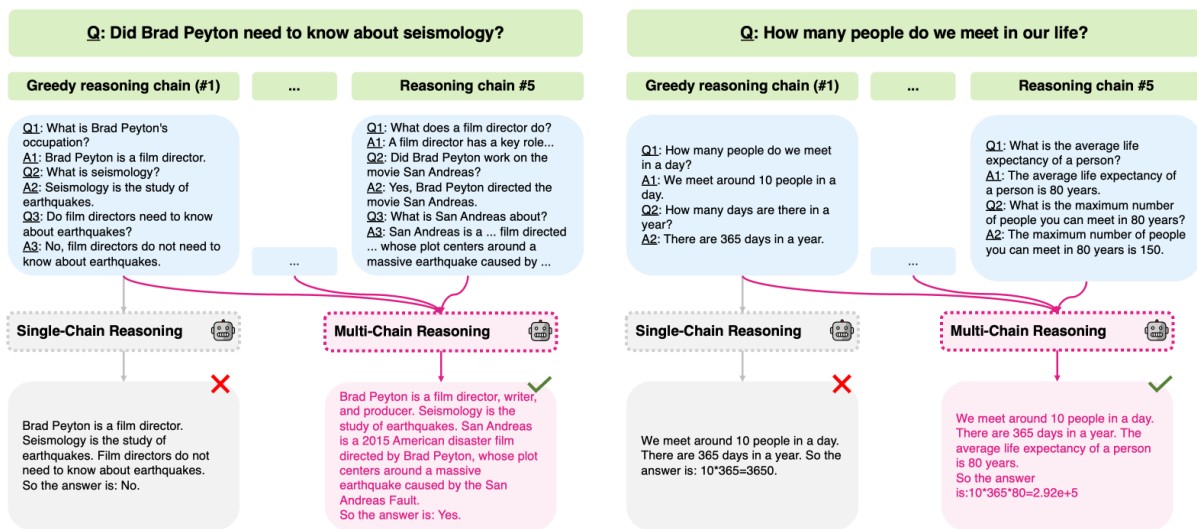

Figure 10: Examples for combining facts from multiple reasoning chains.

| dataset | Reasoning | %3 | %2 | %1 | Sim_predictions | Human_acc | MCR_acc |
|---|---|---|---|---|---|---|---|
| STRATEGYQA | | 79 | 18 | 3 | 89.9 | 77.2 | 72.2 |
| FERMI | implicit | 60 | 32 | 8 | 76.0 | 47.8 | 40.7 |
| QUARTZ | | 91 | 7 | 2 | 98.9 | 87.9 | 86.8 |
| HOTPOTQA | | 77 | 21 | 2 | 95.4 | 49.2 | 49.2 |
| 2WIKIMQA | explicit | 97 | 2 | 1 | 94.9 | 69.6 | 69.2 |
| BAMBOOGLE | | 90 | 9 | 1 | 98.2 | 71.5 | 71.4 |
| Average | implicit | 76.7 | 19.0 | 4.3 | 88.3 | 71.0 | 66.6 |
| Average | explicit | 88.0 | 10.7 | 1.3 | 96.2 | 63.4 | 63.3 |
| Average | | 82.3 | 14.8 | 2.8 | 92.2 | 67.2 | 64.9 |

Table 10: Full results for the explanation quality analysis. *Sim_predictions* indicates the similarity between the human and MCR prediction, calculated using the dataset-specific metrics described in §4.1.1. *Human_acc* and MCR_*acc* represent the accuracy of humans and MCR predictions, respectively. Since only explanations with a score of 3 are guaranteed to contain the necessary information to arrive at an answer, we filter other examples when calculating *sim_predictions*, *Human_acc*, and MCR_*acc*.

| Dataset | SA | SCR | MCR | SCR-EV | MCR-EV |
|---|---|---|---|---|---|
| STRATEGYQA | 10 | 6 | 6 | 6 | 6 |
| FERMI | 6 | 6 | 6 | 6 | 6 |
| QUARTZ | 6 | 8 | 8 | 8 | 8 |
| HOTPOTQA | 12 | 10 | 10 | 10 | 10 |
| 2WIKIMQA | 6 | 6 | 6 | 6 | 6 |
| BAMBOOGLE | 6 | 6 | 6 | 6 | 6 |
| FEVEROUS | 10 | 10 | 10 | 10 | 10 |

Table 11: The number of exemplars for each model and dataset. Since MCR and SCR, and MCR-EV and SCR-EV use the same prompt they have the same number of exemplars.

| Dataset | Dec. | Dec. steps | Dec. out. | Ret. len. | Meta-reason | SCR | MCR |
|---|---|---|---|---|---|---|---|
| STRATEGYQA | 2,242 | 2.9±0.6 | 103.3± 48.0 | 190.6±66.7 | 1,652 | 1,749.0± 52.0 | 2,032.9±110.7 |
| FERMI | 1,442 | 2.3±0.9 | 91.4±31.9 | 165.8±78.9 | 1,681 | 1,765.5±25.5 | 1,984.1±79.6 |
| QUARTZ | 839 | 1.2±0.5 | 55.2±20.6 | 92.7±28.7 | 2,129 | 2,202.4±19.8 | 2,343.6±48.3 |
| HOTPOTQA | 2,508 | 1.7±0.7 | 86.1±91.9 | 153.0±84.1 | 2,380 | 2,460.3±28.1 | 2,666.2±90.0 |
| 2WIKIMQA | 1,920 | 2.4±0.8 | 92.7±30.5 | 201.3±59.3 | 2,029 | 2,116.4±25.6 | 2,363.0±104.6 |
| BAMBOOGLE | 1,342 | 2.0±0.3 | 74.3±37.5 | 204.5±72.1 | 966 | 1,035.1±12.9 | 1,223.1±50.2 |
| FEVEROUS | 3,741 | 2.9±0.9 | 118.2±36.4 | 197.1±69.2 | 2,826 | 2,956.8±38.1 | 3,276.8±123.1 |
| Average | 2,004.9 | 2.2±0.6 | 88.7±18.7 | 172.1±37.0 | 1,951.9 | 2,040.8±563.1 | 2,270.0±587.7 |

Table 12: Prompt lengths (number of tokens) used for each dataset: decomposition prompts (Dec.); number of output decomposition steps (Dec. steps); output decomposition length (Dec. out.); retrieved evidence length (Ret. len.); meta-reasoning prompt; SCR prompt length; MCR prompt length.

| Dataset | Examples | SA | SC@5 | SCR | MCR |
|---|---|---|---|---|---|
| STRATEGYQA | 500 | 66.6 (69.3±0.3) | 71.0 (72.2±0.8) | 68.2 (70.0±0.6) | **72.0 (73.6±0.7)** |
| HOTPOTQA | 500 | 54.3 (50.2±0.3) | 56.2 (51.3±0.2) | 57.1 (56.4±0.4) | **59.3 (57.0±0.8)** |

Table 13: Experiments with `code-davinci-002` when using prompts with different random exemplars for both the decomposition and meta-reasoning prompts. Original prompt results are in parenthesis, for comparison.

Figure 11: The annotation instructions for the MCR explanation quality analysis.

---

*Given a question and a context, answer the question step-by-step. If you are unsure, answer Unknown.*

---

**Context:**
Who is the father of modern Hebrew? The father of modern Hebrew is Eliezer Ben-Yehuda.
Who is the father of Eliezer Ben-Yehuda? The father of Eliezer Ben-Yehuda is Abraham.
...
Who is the father of modern Hebrew? The father of modern Hebrew is Eliezer Ben-Yehuda.
Who is the father of Eliezer Ben-Yehuda? Eliezer Ben-Yehuda's father is Yehuda Leib.

---

**Question:** Who is the father of the father of modern Hebrew?

---

**Answer:** The father of modern Hebrew is Eliezer Ben-Yehuda. The father of Eliezer Ben-Yehuda is Abraham.
**So the answer is:** Abraham.

---

**Gold answer is:** *Yehuda Leib*

---

Figure 12: Example a *Contradicting Facts* error. When generating the explanation, the meta-reasoner has to rely on knowledge encoded in its parameters to decide between multiple contradicting facts in its context on who was the father of Eliezer Ben-Yehuda.

*Given the following question, answer it by providing follow up questions and intermediate answers. If no follow up questions are necessary, answer the question directly. You are also provided with the most relevant google snippet for each intermediate question.*

#

Context1: Xawery Żuławski: Polish-Russian War (Wojna polsko-ruska) is a 2009 Polish film directed by Xawery Żuławski based on the novel Polish-Russian War under the white-red flag by Dorota Masłowska. So the answer is Xawery Żuławski.

Context2: Xawery Żuławski: Xawery Żuławski ; National Film School in Łódź · 1995–present · Maria Strzelecka · 2.

Question: Who is the mother of the director of film Polish-Russian War (Film)?

Are follow up questions needed here: Yes.

Follow up: Who is the director of the film Polish-Russian War (Film)?

Intermediate answer: The director of the film Polish-Russian War is Xawery Żuławski.

Follow up: Who is the mother of Xawery Żuławski?

Intermediate answer: The mother of Xawery Żuławski is Małgorzata Braunek.

So the final answer is: Rick Scott Małgorzata Braunek.

...

#

Context1: John, Count Palatine of Neumarkt: John (Johann von Pfalz-Neumarkt; 1383 – 14 March 1443) was the Count Palatine of Neumarkt from 1410 to his death. The son of Rupert III of the Palatinate, he married Catherine of Pomerania in 1407.

Context2: John, Count Palatine of Neumarkt: John (Johann von Pfalz-Neumarkt; 1383 – 14 March 1443) was the Count Palatine of Neumarkt from 1410 to his death. The son of Rupert III of the Palatinate, he married Catherine of Pomerania in 1407.

Question: Who is Catherine Of Pomerania, Countess Palatine Of Neumarkt's father-in-law?

Are follow up questions needed here: Yes.

Follow up: Who is the husband of Catherine of Pomerania, Countess Palatine of Neumarkt?

Intermediate answer: The husband of Catherine of Pomerania, Countess Palatine of Neumarkt is John, Count Palatine of Neumarkt.

Follow up: Who is the father of John, Count Palatine of Neumarkt?

Intermediate answer: The father of John, Count Palatine of Neumarkt is Rupert III of the Palatinate.

So the final answer is: Rupert III of the Palatinate.

#

Context1: Crimen a las tres: Crimen a las tres is a 1935 Argentine crime film directed and written by Luis Saslavsky. Crimen a las tres. Directed by, Luis Saslavsky.

Context2: Elio Petri: The Working Class Goes to Heaven (Italian: La classe operaia va in paradiso), released in the US as Lulu the Tool, is a 1971 political drama film directed by Elio Petri. So the answer is Elio Petri.

Context3: March 20, 1995: Luis Saslavsky (April 21, 1903 – March 20, 1995) was an Argentine film director, screenwriter and film producer, and one of the influential directors in the Cinema of Argentina of the classic era. So the answer is March 20, 1995.

Context4: Elio Petri: Final years. In 1981, Petri visited Geneva to direct Arthur Millerś new play The American Clock, with Marcello Mastroianni playing the lead role. Petri died of cancer on 10 November 1982. He was 53 years old.

Question: Which film has the director died first, Crimen A Las Tres or The Working Class Goes To Heaven?

Are follow up questions needed here: Yes.

Follow up: Who is the director of Crimen a las tres?

Intermediate answer: The director of Crimen a las tres is Luis Saslavsky.

Follow up: Who is the director of The Working Class Goes to Heaven?

Intermediate answer: The director of The Working Class Goes to Heaven is Elio Petri.

Follow up: When did Luis Saslavsky die?

Intermediate answer: Luis Saslavsky died on March 20, 1995.

Follow up: When did Elio Petri die?

Intermediate answer: Elio Petri died on 10 November 1982.

So the final answer is: The Working Class Goes to Heaven.

#

Figure 13: Instruction and exemplars for the 2WIKIMQA decomposition prompt.

\#

Context1: Frost: Frost is a thin layer of ice on a solid surface, which forms from water vapor in an above-freezing atmosphere coming in contact with a solid surface whose ...

Context2: Graduation: Graduation is the awarding of a diploma to a student by an educational institution. It may also refer to the ceremony that is associated with it.

Context3: Winter: Winter ; Astronomical season, 22 December – 21 March ; Meteorological season, 1 December – 28/29 February ; Solar (Celtic) season, 1 November – 31 January.

Question: Is it common to see frost during some college commencements?

Are follow up questions needed here: Yes.

Follow up: What seasons can you expect to see frost?

Intermediate answer: Frost is common during the winter.

Follow up: When is college commencement?

Intermediate answer: College commencement ceremonies often happen during the months of December, May, June.

Follow up: Do any of the months December, May, June occur during the Winter?

Intermediate answer: December is in the winter.

So the final answer is: Yes.

...

\#

Context1: Last rites: The last rites, also known as the Commendation of the Dying, are the last prayers and ministrations given to an individual of Christian faith, when possible, shortly before death. They may be administered to those awaiting execution, mortally injured, or terminally ill.

Context2: Richard Dawkins: Dawkins is an outspoken atheist and a supporter of various atheist, secular, and humanist organisations, including Humanists UK and the Brights movement. Dawkins suggests that atheists should be proud, not apologetic, stressing that atheism is evidence of a healthy, independent mind.

Context3: Prayer in the Catholic Church: In the Catholic Church, prayer is "the raising of oneś mind and heart to God or the requesting of good things from God." It is an act of the moral virtue ...

Question: Would Richard Dawkins hypothetically refuse an offering of the Last rites?

Are follow up questions needed here: Yes.

Follow up: What are the last Rites?

Intermediate answer: The Last rites, in Catholicism, are the last prayers and ministrations given to an individual of the faith, when possible, shortly before death.

Follow up: What are Richard Dawkins religious beliefs?

Intermediate answer: Richard Dawkins is known as an outspoken atheist, well known for his criticism of creationism and intelligent design.

Follow up: Would an atheist participate in Catholics prayers?

Intermediate answer: It is unlikely that an atheist would participate in Catholics prayers.

So the final answer is: Yes.

\#

Context1: number 1: Hydrogen is the chemical element with the symbol H and atomic number 1. Hydrogen is the lightest element. So the answer is number 1.

Context2: Spice Girls - Simple English Wikipedia, the free encyclopedia: The group has five members. Each member uses a nickname initially given to them: Melanie Chisholm ("Sporty Spice"), Emma Bunton ("Baby Spice"), Melanie Brown ("Scary Spice"), Victoria Beckham (née Adams) ("Posh Spice"), and Geri Halliwell ("Ginger Spice") .

Context3: Square number: In mathematics, a square number or perfect square is an integer that is the square of an integer; in other words, it is the product of some integer with ...

Question: Hydrogen's atomic number squared exceeds number of Spice Girls?

Are follow up questions needed here: Yes.

Follow up: What is the atomic number of hydrogen?

Intermediate answer: Hydrogen has an atomic number of 1.

Follow up: How many people are in the Spice Girls band?

Intermediate answer: The Spice Girls has 5 members.

Follow up: Is the square of 1 greater than 5?

Intermediate answer: The square of 1 is 1 which is less than 5.

So the final answer is: No.

\#

Figure 14: Instruction and exemplars for the STRATEGYQA decomposition prompt.

*Given a question and a context, answer the question and explain why. If you are unsure, answer Unknown.*

#
Context:
Who is the wife of Douglas Leiterman? The wife of Douglas Leiterman is Beryl Fox.
Where was Beryl Fox born? Beryl Fox was born in Winnipeg, Manitoba.
When and where was Mary born? Mary was born in c. 18 BC or September 8 (21), 16 BC Herodian Kingdom of Judea.
Where was Beryl Fox born? Beryl Fox was born in Winnipeg, Manitoba.
Who is the wife of Douglas Leiterman? The wife of Douglas Leiterman is Mary.
Who is the wife of Douglas Leiterman? The wife of Douglas Leiterman is Beryl Fox.

Question:
Where was the wife of Douglas Leiterman born?

Answer:
The wife of Douglas Leiterman is Beryl Fox. Beryl Fox was born in Winnipeg, Manitoba.
So the answer is: Winnipeg, Manitoba.
...
#
Context:
Who is Beatrice of Aragon's father? The father of Beatrice of Aragon is King Ferdinand I of Naples.
Who is the father of Rupert III, Elector Palatine? The father of Rupert III, Elector Palatine is Rupert II, Elector Palatine.
Who is the husband of Catherine of Pomerania? The husband of Catherine of Pomerania is John II, Count Palatine of Neumarkt.
Who is Catherine Of Pomerania, Countess Palatine Of Neumarkt's husband? The husband of Catherine Of Pomerania, Countess Palatine Of Neumarkt is John I, Count Palatine of Neumarkt.
Who is the father of John II, Count of Holstein-Rendsburg? The father of John II, Count of Holstein-Rendsburg is Henry II, Count of Holstein-Rendsburg.
Who is Catherine Of Pomerania, Countess Palatine Of Neumarkt's husband? The husband of Catherine Of Pomerania, Countess Palatine Of Neumarkt is John II, Count of Holstein-Rendsburg.
Who is the father of John I, Count Palatine of Neumarkt? The father of John I, Count Palatine of Neumarkt is Rupert III, Elector Palatine.
Who are the parents of Rupert III, Elector Palatine? The parents of Rupert III, Elector Palatine are Rupert II, Elector Palatine and Beatrice of Aragon.
Who is the father of John II, Count Palatine of Neumarkt? The father of John II, Count Palatine of Neumarkt is Rupert III, Elector Palatine.

Question:
Who is Catherine Of Pomerania, Countess Palatine Of Neumarkt's father-in-law?

Answer:
The husband of Catherine Of Pomerania, Countess Palatine Of Neumarkt is John I, Count Palatine of Neumarkt. The father of John I, Count Palatine of Neumarkt is Rupert III, Elector Palatine.
So the answer is: Rupert III, Elector Palatine.
#
Context:
When did Elio Petri die? Elio Petri died on 10 November 1982.
Who is the director of The Working Class Goes to Heaven? The director of The Working Class Goes to Heaven is Elio Petri.
Who is the director of Crimen A Las Tres? The director of Crimen A Las Tres is Luis Saslavsky.
Who is the director of Crimen A Las Tres? The director of Crimen A Las Tres is Luis Saslavsky.
When did Luis Saslavsky die? Luis Saslavsky died on March 20, 1995.
Who is the director of Crimen A Las Tres? The director of Crimen A Las Tres is Luis Saslavsky.
When did Elio Petri die? Elio Petri died on 10 November 1982.
When did Luis Saslavsky die? Luis Saslavsky died on March 20, 1995.
When did Luis Saslavsky die? Luis Saslavsky died on March 20, 1995.
When did Elio Petri die? Elio Petri died on 10 November 1982.
Who is the director of The Working Class Goes to Heaven? The director of The Working Class Goes to Heaven is Elio Petri.
Who is the director of The Working Class Goes to Heaven? The director of The Working Class Goes to Heaven is Elio Petri.

Question:
Which film has the director died first, Crimen A Las Tres or The Working Class Goes To Heaven?

Answer:
The director of Crimen A Las Tres is Luis Saslavsky. The director of The Working Class Goes to Heaven is Elio Petri. Luis Saslavsky died on March 20, 1995. Elio Petri died on 10 November 1982.
So the answer is: The Working Class Goes To Heaven.
#

Figure 15: Instruction and exemplars for the 2WIKIMQA meta-reasoner prompt.

***Given a question and a context, provide a Yes or No answer and explain why. If you are unsure, answer Unknown.***

#
Context:
What type of animal is a jellyfish? A jellyfish is a type of invertebrate.
Do jellyfish have arteries? No, jellyfish do not have arteries. They have a type of nervous system called a nerve net.
What is atherosclerosis? Atherosclerosis is a disease where plaque builds up in the arteries.
Do jellyfish have arteries? Jellyfish do not have a circulatory system and therefore do not have arteries.
So jellyfish don't have atherosclerosis because they don't have arteries? Yes, that is correct.
What is atherosclerosis? Atherosclerosis is a type of heart disease.
Is an invertebrate susceptible to atherosclerosis? No, invertebrates are not susceptible to atherosclerosis.
What is atherosclerosis? Atherosclerosis is a disease in which plaque builds up on the walls of the arteries.

Question:
Is a jellyfish safe from atherosclerosis?

Answer:
Jellyfish do not have a circulatory system and therefore do not have arteries. Atherosclerosis is a disease in which plaque builds up on the walls of the arteries.
So the answer is: Yes.
#
...
Context:
What is tonsure? Tonsure is the practice of cutting or shaving the hair from the scalp, as a sign of religious devotion or humility.
What is Christopher Hitchens' religious affiliation? Christopher Hitchens was an atheist.
What is tonsure? Tonsure is a practice of cutting or shaving the hair from the scalp.
Would Christopher Hitchens be likely to do something as a sign of religious devotion? No.
Would Christopher Hitchens be religious? Christopher Hitchens was an atheist.
Is an atheist likely to engage in religious practices? No, an atheist is not likely to engage in religious practices.
What is tonsure? Tonsure is the practice of cutting or shaving the hair from the scalp, as a sign of religious devotion or humility.
What is Christopher Hitchens' religious affiliation? Christopher Hitchens was an atheist.
Do atheists usually engage in tonsure? No, atheists do not usually engage in tonsure as it is a religious practice.

Question:
Would Christopher Hitchens be very unlikely to engage in tonsure?

Answer:
Tonsure is the practice of cutting or shaving the hair from the scalp, as a sign of religious devotion or humility. Christopher Hitchens was an atheist. An atheist is not likely to engage in religious practices.
So the answer is: Yes.
#
Context:
What is the evidence that Neanderthals used arithmetic? There is no direct evidence that Neanderthals used arithmetic, but indirect evidence suggests that they may have had some understanding of basic mathematical concepts.
What is the earliest evidence of arithmetic? The earliest evidence of arithmetic dates back to the ancient Sumerians, who lived in what is now modern-day Iraq.
Did Neanderthals live before the ancient Sumerians? Yes.
What is arithmetic? Arithmetic is the branch of mathematics that deals with the properties and manipulation of numbers.
Is there any reason to believe that Neanderthals could not have used arithmetic? There is no reason to believe that Neanderthals could not have used arithmetic.
Do we have any evidence that Neanderthals could manipulate numbers? We have no evidence that Neanderthals could manipulate numbers.
What is arithmetic? Arithmetic is the branch of mathematics that deals with the properties and manipulation of numbers.
What is arithmetic? Arithmetic is the branch of mathematics that deals with the addition, subtraction, multiplication, and division of whole numbers and with the manipulation of fractions and decimals.

Question:
Did Neanderthals use arithmetic?

Answer:
There is no evidence that Neanderthals used arithmetic. The earliest evidence of arithmetic dates back to the ancient Sumerians who lived after the Neanderthals.
So the answer is: No.
#

Figure 16: Instruction and exemplars for the STRATEGYQA meta-reasoner prompt.

*Given a question and a context, answer the question and explain why. If you are unsure, answer Unknown.*

\#
Context:
Who is the wife of Douglas Leiterman? | Douglas Leiterman (1927 – 19 December 2012) was a Canadian television producer.: [[D̂ouglas Leiterman]̂, [Śpouse, B́eryl Fox]̂]
Where was Beryl Fox born? | Winnipeg, Manitoba: Fox was born in 1931 in Winnipeg, Manitoba. So the answer is Winnipeg, Manitoba.
Who is Douglas Leiterman's wife? | Douglas Leiterman (1927 – 19 December 2012) was a Canadian television producer.: [[D̂ouglas Leiterman]̂, [Ḱnown for, T́his Hour Has Seven Days]̂, [Śpouse, B́eryl Fox]̂]
Where was Beryl Fox born? | Winnipeg, Manitoba: Fox was born in 1931 in Winnipeg, Manitoba. So the answer is Winnipeg, Manitoba.
Who is the wife of Douglas Leiterman? | Douglas Leiterman (1927 – 19 December 2012) was a Canadian television producer.: [[D̂ouglas Leiterman]̂, [Śpouse, B́eryl Fox]̂]
When and where was Mary born? | Mary was born in c. 18 BC or September 8 (21), 16 BC Herodian Kingdom of Judea.

Question:
Where was the wife of Douglas Leiterman born?

Answer:
The wife of Douglas Leiterman is Beryl Fox. Beryl Fox was born in Winnipeg, Manitoba.
So the answer is: Winnipeg, Manitoba.
...
\#
Context:
Who is the director of Crimen A Las Tres? | Crimen a las tres: Crimen a las tres is a 1935 Argentine crime film directed and written by Luis Saslavsky.
Who is the director of The Working Class Goes to Heaven? | The Working Class Goes to Heaven: The Working Class Goes to Heaven (Italian: La classe operaia va in paradiso), released in the US as Lulu the Tool, is a 1971 political drama film directed by Elio Petri.
When did Luis Saslavsky die? | Luis Saslavsky: Luis Saslavsky (April 21, 1903 – March 20, 1995) was an Argentine film director, screenwriter and film producer, and one of the influential directors in the Cinema of Argentina of the classic era.
When did Elio Petri die? | Elio Petri: Petri died of cancer on 10 November 1982. He was 53 years old.
Who is the director of Crimen A Las Tres? | Crimen a las tres: Crimen a las tres is a 1935 Argentine crime film directed and written by Luis Saslavsky.
Who is the director of The Working Class Goes to Heaven? | The Working Class Goes to Heaven: The Working Class Goes to Heaven (Italian: La classe operaia va in paradiso), released in the US as Lulu the Tool, is a 1971 political drama film directed by Elio Petri.
When did Luis Saslavsky die? | Luis Saslavsky: Luis Saslavsky (April 21, 1903 – March 20, 1995) was an Argentine film director, screenwriter and film producer, and one of the influential directors in the Cinema of Argentina of the classic era.
When did Elio Petri die? | Elio Petri: Petri died of cancer on 10 November 1982. He was 53 years old.
Who is the director of Crimen A Las Tres? | Crimen a las tres: Crimen a las tres is a 1935 Argentine crime film directed and written by Luis Saslavsky.
When did Luis Saslavsky die? | Luis Saslavsky: Luis Saslavsky (April 21, 1903 – March 20, 1995) was an Argentine film director, screenwriter and film producer, and one of the influential directors in the Cinema of Argentina of the classic era.
Who is the director of The Working Class Goes to Heaven? | The Working Class Goes to Heaven: The Working Class Goes to Heaven (Italian: La classe operaia va in paradiso), released in the US as Lulu the Tool, is a 1971 political drama film directed by Elio Petri.
When did Elio Petri die? | Elio Petri: Petri died of cancer on 10 November 1982. He was 53 years old.

Question:
Which film has the director died first, Crimen A Las Tres or The Working Class Goes To Heaven?

Answer:
The director of Crimen A Las Tres is Luis Saslavsky. The director of The Working Class Goes to Heaven is Elio Petri. Luis Saslavsky died on March 20, 1995. Elio Petri died on 10 November 1982.
So the answer is: The Working Class Goes To Heaven.
\#

Figure 17: Instruction and exemplars for the 2WⁱᴋⁱMQA meta-reasoner prompt for MCR-Eᴠ and SCR-Eᴠ reasoning over retrieved evidence.

*Given a question and a context, answer the question step-by-step. If you are unsure, answer Unknown.*

#
Context:
What is atherosclerosis? | Atherosclerosis: Atherosclerosis is a pattern of the disease arteriosclerosis in which the wall of the artery develops abnormalities, called lesions. These lesions may lead to narrowing due to the buildup of atheromatous plaque. At onset there are usually no symptoms, but if they develop, symptoms generally begin around middle age.
What type of animal is a jellyfish? | Jellyfish - Simple English Wikipedia, the free encyclopedia: Jellyfish are animals of the phylum Cnidaria. They are a monophyletic clade, the Medusozoa. Most of them live in the oceans, in salt water, where they eat small sea animals like plankton and little fish, and float in the sea.
Is an invertebrate susceptible to atherosclerosis? | Atherosclerosis: Atherosclerosis is a pattern of the disease arteriosclerosis in which the wall of the artery develops abnormalities, called lesions.
What is atherosclerosis? | Atherosclerosis: Atherosclerosis is a pattern of the disease arteriosclerosis in which the wall of the artery develops abnormalities, called lesions. These lesions may lead to narrowing due to the buildup of atheromatous plaque. At onset there are usually no symptoms, but if they develop, symptoms generally begin around middle age.
Do jellyfish have arteries? | Jellyfish: Jellyfish are mainly free-swimming marine animals with umbrella-shaped bells and trailing tentacles, although a few are anchored to the seabed by stalks rather
What is atherosclerosis? | Atherosclerosis: Atherosclerosis is a pattern of the disease arteriosclerosis in which the wall of the artery develops abnormalities, called lesions. These lesions may lead to narrowing due to the buildup of atheromatous plaque. At onset there are usually no symptoms, but if they develop, symptoms generally begin around middle age.
Do jellyfish have arteries? | Jellyfish: Jellyfish are mainly free-swimming marine animals with umbrella-shaped bells and trailing tentacles, although a few are anchored to the seabed by stalks rather
So jellyfish don't have atherosclerosis because they don't have arteries? | Jellyfish: A free-swimming marine coelenterate that is the sexually reproducing form of a hydrozoan or scyphozoan and has a nearly transparent saucer-shaped body and

Question:

Is a jellyfish safe from atherosclerosis?
Answer:
Jellyfish do not have a circulatory system and therefore do not have arteries. Atherosclerosis is a disease in which plaque builds up on the walls of the arteries.
So the answer is: Yes.
...
#
Context:
What is arithmetic? | Arithmetic: Arithmetic is an elementary part of mathematics that consists of the study of the properties of the traditional operations on numbers—addition, subtraction, multiplication, division, exponentiation, and extraction of roots.
What is the evidence that Neanderthals used arithmetic? | Neanderthal: In 2012, British-American geneticist Graham Coop hypothesised that they instead found evidence of a different archaic human species interbreeding with modern
Is there any reason to believe that Neanderthals could not have used arithmetic? | Neanderthal: A large part of the controversy stems from the vagueness of the term "species", as it is generally used to distinguish two genetically isolated populations, but
What is arithmetic? | Arithmetic: Arithmetic is an elementary part of mathematics that consists of the study of the properties of the traditional operations on numbers—addition, subtraction, multiplication, division, exponentiation, and extraction of roots.
Do we have any evidence that Neanderthals could manipulate numbers? | Neanderthal: Neanderthals also written as Neandertals, are an extinct species or subspecies of archaic humans who lived in Eurasia until about 40,000 years ago.
What is arithmetic? | Neanderthal: Neanderthals also written as Neandertals, are an extinct species or subspecies of archaic humans who lived in Eurasia until about 40,000 years ago.
What is the earliest evidence of arithmetic? | Mathematics: It is in Babylonian mathematics that elementary arithmetic (addition, subtraction, multiplication, and division) first appear in the archaeological record. The Babylonians also possessed a place-value system and used a sexagesimal numeral system which is still in use today for measuring angles and time.
Did Neanderthals live before the ancient Babylonians? | Neanderthal: Neanderthals also written as Neandertals, are an extinct species or subspecies of archaic humans who lived in Eurasia until about 40,000 years ago. Pre- and early Neanderthals, living before the Eemian interglacial

Question:
Did Neanderthals use arithmetic?

Answer:
There is no evidence that Neanderthals used arithmetic. The earliest evidence of arithmetic dates back to the ancient Babylonians who lived after the Neanderthals.
So the answer is: No.
#

Figure 18: Instruction and exemplars for the STRATEGYQA prompt for MCR-Ev and SCR-Ev reasoning over retrieved evidence.