# OpenReview forum: "Answering Questions by Meta-Reasoning over Multiple Chains of Thought"
_EMNLP/2023/Conference — EMNLP 2023 Main_

### Official Review · Reviewer_hqxw · 2023-07-30

**Soundness:** 4

**Excitement:**

3: Ambivalent: It has merits (e.g., it reports state-of-the-art results, the idea is nice), but there are key weaknesses (e.g., it describes incremental work), and it can significantly benefit from another round of revision. However, I won't object to accepting it if my co-reviewers champion it.

**Paper Topic And Main Contributions:**

This paper proposes a meta-reasoning framework to improve the reasoning capability of LLM on multi-hop QA tasks. The meta-reasoning reasons among multiple reasoning chains generated by CoT prompting. Since the reasoning chains generated by CoT method may provide some hallucinated evidences that cause an incorrect answer, the proposed meta-reasoning is capable to pick up the correct evidences from multiple reasoning chains to make a reliable answer generation. To realize meta-reasoning, the reasoning chains for a question generated from an LLM are fed into another LLM to conduct another round of CoT reasoning before generating the answer.

**Questions For The Authors:**

1. Could you make statistics of your method and all baseline models about the entire length of the input sequence, including all the intermediate prompts, evidences, and reasoning chains?
2. What is the performance of the retriever in retrieving evidence facts? If the retriever fails to retrieve relevant evidences, will it heavily hurt the final QA performance?

**Reasons To Accept:**

1. The motivation of this paper is clear. That is, a second-round of reasoning to pick the most relevant evidences from multiple reasoning chains.

**Reasons To Reject:**

1. The method is not novel. Based on my understanding, you just prompt the LLM with a second-round CoT, additionally including the multiple reasoning chains sampled from the first-round CoT. It is more like an engineering practice. At least, I did not find this work to be exciting.
2. The improvements compared to baselines are marginal.
3. Based on my understanding, the method is computationally intensive and not efficient. The input text is very long and it needs to trigger LLMs twice, one for CoT and one for meta-reasoning.

**Reproducibility:**

4: Could mostly reproduce the results, but there may be some variation because of sample variance or minor variations in their interpretation of the protocol or method.

**Reviewer Confidence:**

3: Pretty sure, but there's a chance I missed something. Although I have a good feel for this area in general, I did not carefully check the paper's details, e.g., the math, experimental design, or novelty.

---

> ### Author Rebuttal · Authors · 2023-08-28
>
> We thank the reviewer for the positive comment regarding the motivation of the paper. We will now address all relevant points and questions:
>
>
> *P1. The method is not novel. Based on my understanding, you just prompt the LLM with a second-round CoT, additionally including the multiple reasoning chains sampled from the first-round CoT. It is more like an engineering practice. At least, I did not find this work to be exciting.*
>
>
> P1.A. Leveraging multiple CoTs is an active area of research [1,2], and to our knowledge meta-reasoning is **the first method to use a LLM to review multiple reasoning strategies**. Aside from aggregating multiple CoTs, meta-reasoning is closely tied to self-reflection in LLMs, which is becoming increasingly important [3,4] (contemporaneous work). While it is discussed in our related work (541-546), we will be sure to further emphasize this in the camera-ready version. Second, while the idea of meta-reasoning is simple, there is merit in its simplicity, as it is a general solution that is applicable to multiple combinations of LLMs, retrievers and QA tasks.
>
> [1] Self-Consistency Improves Chain of Thought Reasoning in Language Model;
> [2] Tree of Thoughts: Deliberate Problem Solving with Large Language Models;
> [3] Reflexion: an autonomous agent with dynamic memory and self-reflection;
> [4] Self-refine: Iterative refinement with self-feedback
>
>
> *P2. The improvements compared to baselines are marginal.*
>
>
> P2.A. Our approach demonstrates **consistent improvements on 7 different QA tasks**, when compared to strong baselines (self-ask, self-consistency). These gains are maintained across different retrievers (ColBERTv2, Google) and LLMs (code-davinci-002, Vicuna-13B), as shown in Tables 2-4, 7-8. Furthermore, MCR generates quality explanations, which serve as an additional advantage over self-consistency. We believe that these gains demonstrate the advantage of MCR, compared to existing approaches.
>
>
> *P3. Based on my understanding, the method is computationally intensive and not efficient. The input text is very long and it needs to trigger LLMs twice, one for CoT and one for meta-reasoning.*
>
>
> P3.A. In the paper, we compare MCR to self-consistency (SC) and show, in lines 366-8, that it is in fact **more efficient** than SC, achieving higher performance (on 6/7 tasks) while using much fewer LLM calls (sampling 5 chains + 1 call to the meta-reasoner for MCR vs sampling 15 chains for SC). We consider SC to be a natural baseline as it is a standard approach for CoT which involves multiple calls to the LLM and has been shown to boost QA performance [1,2,3]. For a comparison of LLM input lengths between MCR and SC see A1 below.
>
> [1] Demonstrate-Search-Predict: Composing retrieval and language models for knowledge-intensive NLP;
> [2] Faithful Chain-of-Thought Reasoning;
> [3] Large Language Models are Zero-Shot Reasoners
>
>
> *Q1. Could you make statistics of your method and all baseline models about the entire length of the input sequence, including all the intermediate prompts, evidences, and reasoning chains?*
>
>
> A1. We provide below all the relevant statistics that were requested: input sequence lengths including all the intermediate prompts, evidence, and reasoning chains. Stats are provided for our SA baseline (also used by SC and MCR), as well as for the meta-reasoning models. We would be happy to include these in our camera-ready version. We will also make our prompts, data, and code publicly available as mentioned in the paper. A few notes:
> - The table below describes the number of tokens for code-davinci-002, matching Table 2.
> - As requested, we provide prompt lengths for all of our baselines: SA, SCR and MCR.
> - Generating a single reasoning chain (i.e. CoT prompting) requires multiple LLM calls, one for each decomposition step. This means that a single decomposition generation is generally longer than applying our additional meta-reasoning step. For more details on our method’s efficiency see P2.A above.
>
> |   Dataset  | SA prompt length | Number of decomposition steps | SA Inference length (avg, std) | Retrieved evidence length (avg, std) | SCR/MCR prompt length (no decompositions) | SCR prompt length (with decompositions) (avg, std) | MCR prompt length (with decompositions) (avg, std) | SCR inference length (avg, std) | MCR inference length (avg, std) |
> |:----------:|:----------------:|:-----------------------------:|:------------------------------:|:------------------------------------:|:-----------------------------------------:|:--------------------------------------------------:|:--------------------------------------------------:|:-------------------------------:|:-------------------------------:|
> | StrategyQA |       2242       |            2.9±0.6            |           103.3± 48.0          |              190.6±66.7              |                    1652                   |                    1749.0± 52.0                    |                    2032.9±110.7                    |            41.9±19.7            |            47.4±20.8            |
> | Fermi      |       1442       |            2.3±0.9            |            91.4±31.9           |              165.8±78.9              |                    1681                   |                     1765.5±25.5                    |                     1984.1±79.6                    |            42.6±17.0            |            46.6±18.1            |
> | Quartz     |        839       |            1.2±0.5            |            55.2±20.6           |               92.7±28.7              |                    2129                   |                     2202.4±19.8                    |                     2343.6±48.3                    |             26.1±7.4            |            36.2±18.1            |
> | HotpotQA   |       2508       |            1.7±0.7            |            86.1±91.9           |              153.0±84.1              |                    2380                   |                     2460.3±28.1                    |                     2666.2±90.0                    |            36.9±12.9            |            39.4±13.2            |
> | 2WikiHop   |       1920       |            2.4±0.8            |            92.7±30.5           |              201.3±59.3              |                    2029                   |                     2116.4±25.6                    |                    2363.0±104.6                    |            46.7±14.8            |            46.9±14.1            |
> | Bamboogle  |       1342       |            2.0±0.3            |            74.3±37.5           |              204.5±72.1              |                    966                    |                     1035.1±12.9                    |                     1223.1±50.2                    |             36.3±7.4            |            37.5±10.6            |
> | Feverous   |       3741       |            2.9±0.9            |           118.2±36.4           |              197.1±69.2              |                    2826                   |                     2956.8±38.1                    |                    3276.8±123.1                    |             14.6±9.6            |             15.4±9.2            |
> | Average    |   2004.9±882.1   |            2.2±0.6            |            88.7±18.7           |              172.1±37.0              |                1951.9±550.3               |                    2040.8±563.1                    |                    2270.0±587.7                   |            35.0±10.3            |            38.5±10.4            |
>
>
> *Q2. What is the performance of the retriever in retrieving evidence facts? If the retriever fails to retrieve relevant evidences, will it heavily hurt the final QA performance?*
>
>
> A2. In the error analysis (Table 5) we list the percentage of errors due to the retriever, for each dataset. Furthermore, we compare the performance of code-davinci-002 with two separate retrievers in Tables 2 and Table 7 (in the appendix). For both of these retrievers, MCR outperforms all other baselines. While the choice of retriever affects decomposition performance, it does so for all of our baselines (SA, SC and MCR). Overall, while retriever performance is important, it is orthogonal to our paper’s contribution which is the meta-reasoning approach.
>
> To explicitly address the reviewer’s questions we performed a manual analysis of 50 random examples from 3 of our datasets, which we would add to our camera-ready version. We reviewed the greedy reasoning chain generated by our retrieval augmented decomposition model. For each chain, we labeled whether or not it contained all of the gold evidence for answering the multi-hop question. This analysis also suggests that good retrieval is necessary to achieve high accuracy.
>
> | Dataset (50 random examples) | Google retriever accuracy  (retrieved all gold evidence, SA greedy) | SA acc. where retriever is correct / incorrect |
> |:----------------------------:|:-------------------------------------------------------------------:|:----------------------------------------------:|
> | Bamboogle                    |                                 68%                                 |                  80.6% / 29.1%                 |
> | 2WikiMQA                     |                                 78%                                 |                  70.4% / 28.8%                 |
> | StrategyQA                   |                                 58%                                 |                  72.4% / 42.9%                 |
>
>
> Note that for StrategyQA retrieval is more challenging, as many questions involve commonsense reasoning which cannot be directly inferred from the retrieved evidence (e.g. the answer to the question “Can jackfruit be used as a weapon?” is “Yes” because it is covered with spikes, however, the retriever fails to retrieve this evidence).
>
>
> ---------------
>
> In conclusion, we have addressed all of the issues raised by the reviewer and will discuss them in our camera-ready version. We are hopeful that, given these clarifications, the reviewer would be willing to raise their Soundness score (and potentially find the work to be more exciting), in line with the other reviewers.

---

### Official Review · Reviewer_reqd · 2023-08-01

**Soundness:** 4

**Excitement:**

4: Strong: This paper deepens the understanding of some phenomenon or lowers the barriers to an existing research direction.

**Paper Topic And Main Contributions:**

This paper proposed Multi Chain Reasoning (MCR), an approach which prompts large language models to meta-reason over multiple chains of thought, rather than aggregate their answers. MCR examines different reasoning chains, mixes information between them and selects the most relevant facts in generating an explanation and predicting the answer. MCR generally contains two components, generating reasoning chains and reasoning over reasoning chains. The reasoning chains is generated by a question decomposition model and a retriever in a iterative way. With multiple reasoning chains generated in the last step, the authors then deploy meta-reasoner module to derive the final answer from multiple chains. All components except the retriever are based on LLMs. They experimented MCR on 7 multihop reasoning datasets and the results show that MCR outperforms all baselines, including self-consistency. They also examined the explanation quality generated by the meta reasoner and found that most of the explanations are faithful.

**Questions For The Authors:**

- How does the performance of MCR vary according to different number of reasoning chains? It would make the results more comprehensive if the authors could draw a figure showing performance vs. number of chains used by MCR.
- How does the temperature affect the final performance?
- The experiments in this paper are conducted only using multihop reasoning datasets. Do you think similar conclusion can be generalized to other domains such as mathematical reasoning?

**Reasons To Accept:**

- To my knowledge, the proposed method to reason over multiple reasoning chains is novel and through comprehensive experiments, the authors demonstrate the effectiveness of the proposed method over strong baselines.
- Although the idea is straightforward, the amount of efforts to make the whole pipeline work is substantial. Personally, I really appreciate the efforts the authors put into to ensure reproducibility of this work. I think this paper could foster future work that focuses on better utilizing multiple reasoning chains sampled from LLMs.
- This paper is well written and presented.

**Reasons To Reject:**

I don't have any specific reasons for rejecting this paper. However, I do have several questions, see below.

**Reproducibility:**

4: Could mostly reproduce the results, but there may be some variation because of sample variance or minor variations in their interpretation of the protocol or method.

**Reviewer Confidence:**

4: Quite sure. I tried to check the important points carefully. It's unlikely, though conceivable, that I missed something that should affect my ratings.

---

> ### Author Rebuttal · Authors · 2023-08-28
>
> We thank the reviewer and are grateful for their appreciation of the reproducibility efforts put into this work. Regarding the reviewer questions:
>
> *Q1.How does the performance of MCR vary according to different number of reasoning chains? It would make the results more comprehensive if the authors could draw a figure showing performance vs. number of chains used by MCR.*
>
> A1. In the paper (Tables 2 and 3), we show that meta-reasoning with 5 reasoning chains (MCR) is better than with 1 chain (SCR) and that MCR can be extended to 15 reasoning chains by applying self-consistency over 3 MCR runs. We will be sure to include the suggested plot in the camera-ready version of our paper. Namely, we will plot the averaged accuracy of MCR as a function of the number reasoning chains used at different data points (n=1,3,5,...,15).
>
>
> *Q2. How does the temperature affect the final performance?*
>
> A2. We empirically observe that greedy decoded reasoning chains (temp=0) display a higher quality than those sampled with temp=0.7. For example, 64.6 vs 58.9 on Bamboogle; 78.3 vs 71.0 on Quartz; 63.8 vs 58.3 on 2WikiMQA. In our setting, we follow past work (e.g self-consistency) and sample multiple reasoning chains using temperature. For MCR we combine the greedy decoded chain with 4 chains sampled with temp=0.7 (lines 298-300). For SC, we use the same chains as MCR (lines 324-326). It could be interesting to explore other sampling methods in future work.
>
>
> *Q3.The experiments in this paper are conducted only using multihop reasoning datasets. Do you think similar conclusion can be generalized to other domains such as mathematical reasoning?*
>
> A3. Our focus in the paper was on open-domain QA, as it is a challenging setting, where there are different strategies to solve a question and different ways to leverage the retrieved evidence. We therefore opted not to experiment with mathematical reasoning datasets in this work. However, we have no reason to believe that MCR should fail on mathematical tasks, in particular those where multiple reasoning chains can be useful.

---

### Official Review · Reviewer_aP2G · 2023-08-04

**Soundness:** 4

**Excitement:**

4: Strong: This paper deepens the understanding of some phenomenon or lowers the barriers to an existing research direction.

**Paper Topic And Main Contributions:**

This work presents a new method for LLMs to perform reasoning. Unlike the previous self-consistency method that just takes the majority vote of multiple reasoning chains as the final answer, the proposed MCR method feeds all the questions and answers from every reasoning chain to another LLM and then uses it to generate the final answer. Following previous works, in the described system, the reasoning chain generation also involves a decomposition model and a retrieval model. The experiments are conducted on 7 different datasets. MCR consistently outperforms self-consistency or at least achieves similar results. When a large number of reasoning chains are available, a combination of MCR and SC can also lead to further improvements. Finally, this work provides a detailed analysis to demonstrate some reasons behind the success of this method, as well as an error analysis.

**Questions For The Authors:**

1. How sensitive is MCR to the choices of prompts and few-shot examples? How long does it take for the authors to design the prompts in the appendix? And in general, will it be easy to design these prompts given a new task with little domain knowledge?
2. Both CoT prompting and self-consistency methods can be used with a variety of reasoning chain formats. In the presented work, all the experiments are conducted with a specific style of reasoning chain (i.e., a sequence of questions, evidence, and answers). I wonder if the authors have tried other styles of reasoning chains (even if the absolute performance is worse than the reported numbers), and what are the performance trends there?

**Reasons To Accept:**

1. This is a straightforward idea leading to nice improvement over self-consistency. The evaluation is comprehensive on seven different datasets, and the authors also show that the combination can achieve further improvements.
2. This paper is well-written. It is very clear what the contribution of this paper is, and what is the similarities and differences compared to previous methods.
3. This work contains a detailed analysis, providing useful insights behind the performance improvements. The error analysis can also be useful for future work.

**Reasons To Reject:**

I have several questions to ask the author about the paper's details, but I don't see a strong reason to reject this paper.

**Reproducibility:**

4: Could mostly reproduce the results, but there may be some variation because of sample variance or minor variations in their interpretation of the protocol or method.

**Reviewer Confidence:**

4: Quite sure. I tried to check the important points carefully. It's unlikely, though conceivable, that I missed something that should affect my ratings.

---

> ### Author Rebuttal · Authors · 2023-08-28
>
> Thank you for the positive comments regarding our evaluation and analysis. Please see our response below:
>
> *Q1: How sensitive is MCR to the choices of prompts and few-shot examples? How long does it take for the authors to design the prompts in the appendix? And in general, will it be easy to design these prompts given a new task with little domain knowledge?*
>
> A1. Designing the prompts did not take long, we tried only a handful of prompts and did not engage in any significant prompt engineering. For our decomposition prompts, we selected random examples from the datasets and annotated them with gold reasoning chains (intermediate questions, answers and evidence). To construct the meta-reasoning prompts, we used random examples, their decomposition-generated reasoning chains as evidence, and manually annotated them with a chain-of-thought explanation that filters the relevant evidence from the generated chains. Therefore, designing these prompts for a new task should be straightforward, given a handful of random examples and their gold decompositions.
>
> Regarding MCR’s sensitivity to the choice of prompt, we added new experiments where we randomly sampled new exemplars for both our decomposition and meta-reasoning prompts for StrategyQA and HotpotQA. When using different random exemplars, we observe that MCR **still outperforms** all baselines. Even though decomposition performance (self-ask) is more affected by the set of exemplars, the performance trend remains the same, with MCR being on top. Below are the experiment results, evaluated on 500 examples from each dataset (paper results are in parenthesis and are averaged over 3 runs with the original prompts).
>
> |   Dataset  | Number of examples | Number of seeds |     Self-Ask    |        SC@5       |       SCR       |       MCR       |
> |:----------:|:------------------:|:---------------:|:---------------:|:-----------------:|:---------------:|:---------------:|
> | StrategyQA |      500 (1000)     |      1 (3)      | 66.6 (69.3±0.3) |  71.0 (72.2±0.8)  | 68.2 (70.0±0.6) | **72.0 (73.6±0.7)** |
> | HotpotQA   |     500 (500)     |      1 (3)      | 54.3 (50.2±0.3) | 56.2 (51.3±0.2) | 57.1 (56.4±0.4) | **59.3 (57.0±0.8)** |
>
>
> *Q2. Both CoT prompting and self-consistency methods can be used with a variety of reasoning chain formats. In the presented work, all the experiments are conducted with a specific style of reasoning chain (i.e., a sequence of questions, evidence, and answers). I wonder if the authors have tried other styles of reasoning chains (even if the absolute performance is worse than the reported numbers), and what are the performance trends there?*
>
> A2. We had preliminary results using an alternative reasoning chain format: a sequence of intermediate questions and answers without any evidence. MCR **still outperformed** SC@5 on the alternative format on StrategyQA (75.5 vs 72.2); Quartz (79.8 vs 77.1); 2WikiMQA (44.2 vs 40.5). However, as our focus was on an open-domain setting, we ultimately opted to incorporate retrieved evidence as well. We view extending MCR to new formats and settings as an interesting topic to explore in future work.

---

### Meta-Review · Area_Chair_MBkL · 2023-09-08

**Recommendation:** 5

**Metareview:**

This paper improves upon prior work that aggregates over multiple chain-of-thought outputs to generate a final answer. The idea is to use all reasoning chains and have the LLM use all of them as context to predict the answer. Experiments on seven datasets confirm the effectiveness of this approach. The reviewers found the paper convincing and well-written.

---

### Decision · Program_Chairs · 2023-10-07

**Decision:**

Accept-Main

**Comment:**

This paper improves upon prior work that aggregates over multiple chain-of-thought outputs to generate a final answer. The idea is to use all reasoning chains and have the LLM use all of them as context to predict the answer. Experiments on seven datasets confirm the effectiveness of this approach. The reviewers found the paper convincing and well-written.